# Crosstalk between CST and RPA regulates RAD51 activity during replication stress

Kai-Hang Lei [1,6], Han-Lin Yang[2,6], Hao-Yen Chang[1], Hsin-Yi Yeh[1], Dinh Duc Nguyen [3], Tzu-Yu Lee [2], Xinxing Lyu[3], Megan Chastain[4], Weihang Chai [3], Hung-Wen Li [2✉] & Peter Chi [1,5✉]

Replication stress causes replication fork stalling, resulting in an accumulation of single-stranded DNA (ssDNA). Replication protein A (RPA) and CTC1-STN1-TEN1 (CST) complex bind ssDNA and are found at stalled forks, where they regulate RAD51 recruitment and foci formation in vivo. Here, we investigate crosstalk between RPA, CST, and RAD51. We show that CST and RPA localize in close proximity in cells. Although CST stably binds to ssDNA with a high affinity at low ionic strength, the interaction becomes more dynamic and enables facilitated dissociation at high ionic strength. CST can coexist with RPA on the same ssDNA and target RAD51 to RPA-coated ssDNA. Notably, whereas RPA-coated ssDNA inhibits RAD51 activity, RAD51 can assemble a functional filament and exhibit strand-exchange activity on CST-coated ssDNA at high ionic strength. Our findings provide mechanistic insights into how CST targets and tethers RAD51 to RPA-coated ssDNA in response to replication stress.

[1] Institute of Biochemical Sciences, National Taiwan University, Taipei, Taiwan. [2] Department of Chemistry, National Taiwan University, Taipei, Taiwan. [3] Department of Cancer Biology, Cardinal Bernardin Cancer Center, Loyola University Chicago Stritch School of Medicine, Maywood, IL, USA. [4] Office of Research, Washington State University, Spokane, WA, USA. [5] Institute of Biological Chemistry, Academia Sinica, Taipei, Taiwan. [6] These authors contributed equally: Kai-Hang Lei, Han-Lin Yang. ✉email: hwli@ntu.edu.tw; peterhchi@ntu.edu.tw

Faithful DNA replication is crucial for the maintenance of genome integrity. The replication machinery encounters many obstacles, including DNA lesions, DNA secondary structure, fragile sites, limiting nucleotides, and RNA/DNA hybrids, all of which may induce replication stress[1]. Under replication stress, continuous progression of the replication fork is often interrupted, leading to stalled replication forks and an accumulation of ssDNA[2]. The heterotrimeric ssDNA-binding protein RPA acts as "first responder" in binding the exposed ssDNA, thereby protecting stressed forks from nuclease attack and helping to activate serine/threonine-specific protein kinase ATR checkpoint signaling[3,4]. The stalled forks then undergo a reversal process that is regulated by RPA, RAD51 and DNA translocases to form a four-way junction structure[5–10]. Replication fork reversal, also known as fork regression, can prevent fork breakage and also temporarily halts the replication process to facilitate fork repair or fork restart[3,10]. However, reversed forks are vulnerable to attack by nucleases such as MRE11, EXO1, and DNA2[11–14]. Consequently, reversed forks must be protected[15].

RAD51, a conserved general recombinase, is a central player in promoting fork reversal and in protecting reversed forks from nucleolytic attack[16–20], and it is responsible for DNA double-strand break (DSB) repair via homologous recombination (HR) by catalyzing DNA strand exchange with a homologous chromatid[21]. In its recombinase role, RAD51 forms a functional nucleoprotein filament on ssDNA that is capable of searching for and locating the homologous template to initiate repair[21,22]. Importantly, association of RAD51 with stressed forks is necessary for fork reversal and for protecting reversed forks from MRE11-mediated degradation. This attribute of RAD51 is independent of its DNA strand-exchange activity. However, the DNA strand-exchange activity of RAD51 is important for restarting replication forks (Fig. 1a)[20]. Assembly of RAD51-ssDNA filaments is a rate-limiting step due to interference by the abundant ssDNA-binding protein RPA, which effectively competes with RAD51 for sites on DNA[23]. To overcome the inhibitory effect of RPA on RAD51-ssDNA filament assembly, accessory factors facilitate RAD51 loading onto RPA-bound ssDNA and stabilize the RAD51 nucleoprotein filament[24–27]. It is worth noting that RAD51 can catalyze the preformed RPA-ssDNA substrate for strand exchange at higher concentrations[23,28,29]. Our recent in situ protein interactions at nascent and stalled replication forks (SIRF) analysis[30] demonstrates that the CST complex facilitates recruitment of RAD51 to stalled forks upon hydroxyurea treatment[31,32].

Mutations in *CTC1* and *STN1* have been found in patients with Coats plus disease, an autosomal recessive disorder resulting in growth retardation, neurological disorder, retinal telangiectasia, bone marrow failure, anemia, graying hair, osteoporosis, and liver fibrosis[33–35]. Certain symptoms of premature aging among Coats plus patients link the evolutionarily conserved role of the CST complex to maintenance of telomere integrity[36]. It has been shown previously that CST facilitates telomere replication, and it coordinates G- and C-strand synthesis by stimulating the priming activity of DNA polymerase α (Polα)-primase complex for C-strand synthesis and by blocking telomerase access to telomeres to prevent excessive G-strand elongation[36–40].

Apart from its role in telomere homeostasis, CST also functions in maintaining genomic integrity[32,41,42]. CST regulates choice of DSB repair pathway in a Shieldin-dependent manner[43,44]. During DNA replication, CST disrupts the interaction of Chromatin licensing and DNA replication factor (CDT1) with the Minichromosome maintenance helicase complex (MCM) to limit replication origin firing and it interacts with AND1/Polα to promote replisome assembly[42]. Notably, CST also facilitates replication fork recovery under replication stress[32,41].

Previous work by us and others has documented that the CST complex participates in reinitiating stalled DNA synthesis at both telomeric and non-telomeric sequences[32,36,37,41,45]. DNA fiber analysis has further evidenced that STN1 or CTC1 depletion enhanced fork degradation and that RNAi-resistant wild-type STN1 or CTC1, respectively, rescued that phenotype[31]. Overall then, the CST complex plays an important role in protecting and restarting stalled replication forks. Importantly, depletion of CST resulted in genome instabilities similar to those displayed by BRCA2-depleted cells[14,31]. Mechanistically, we have provided evidence that CST localizes at stalled replication forks upon hydroxyurea treatment and that CST depletion causes MRE11-mediated degradation of nascent strand DNA at reversed forks, with purified CST complex blocking degradation of DNA by MRE11 in vitro[31].

CST, like RPA, harbors multiple oligonucleotide/oligosaccharide-binding (OB)-fold domains and possesses a high affinity for ssDNA[46,47]. Single-particle reconstruction of CST-ssDNA nucleoprotein complex by cryogenic electron microscopy (cryo-EM) has revealed that it has a decameric configuration[47]. Both RPA and CST are enriched at fragile sites in response to replication fork stalling[32,48,49]. Importantly, CST co-immunoprecipitates with RAD51, and SIRF analysis has shown that CST depletion limits recruitment of RAD51 to stalled forks[31,32].

Here, we aim to establish the mechanism by which CST and RPA influence RAD51-ssDNA nucleoprotein filament assembly. Our cellular imaging analyses reveal that CST is proximal to RPA at stalled replication forks. In support of this notion, our biochemical and biophysical studies reveal that CST and RPA can co-occupy the same ssDNA template. We further show that CST physically interacts with RAD51 and recruits RAD51 to RPA-bound ssDNA. However, CST cannot assist RAD51-mediated DNA strand exchange using RPA-coated ssDNA as substrate. Thus, our results indicate that CST recruits and tethers RAD51 to RPA-coated ssDNA at stalled/collapsed forks during replication stress. Notably, our findings also reveal that although RPA-coated ssDNA inhibits RAD51 activity, RAD51 can assemble a functional filament and exhibit strand-exchange activity on CST-coated ssDNA at high ionic strength. We discuss the physiological relevance of this crosstalk among RPA, CST, and RAD51.

## Results

**CST and RPA localize in close proximity in cells.** During replication stress, stalled replication forks may convert to reversed forks to prevent fork collapse[10]. Assembly of the RAD51 filament is a prerequisite for protecting forks[16,50]. Both CST and RPA are OB-fold-containing protein complexes that dynamically regulate RAD51 filament formation. Our previous study has revealed that CST localizes at stalled forks and that the DNA-binding activity of CST is needed for recruitment of RAD51 to stalled forks in response to hydroxyurea (HU) treatment[31]. Interestingly, both CST and RPA have been shown to localize at GC-rich fragile sites[32,48], raising the intriguing question of how these two ssDNA-binding protein complexes regulate RAD51 filament formation under replication stress (Fig. 1a).

To explore this question, first we examined the spatial relationship of CST and RPA in cells by performing a proximity ligation assay (PLA, Fig. 1b), which is a highly sensitive method for detecting two proteins that are in close proximity in situ[51]. We detected robust PLA signals between endogenous CTC1/STN1 and RPA32, and those signals were significantly enhanced upon HU treatment (Fig. 1c). CTC1/STN1 knockdown limited PLA signal, suggesting that the observed PLA signals were specific to

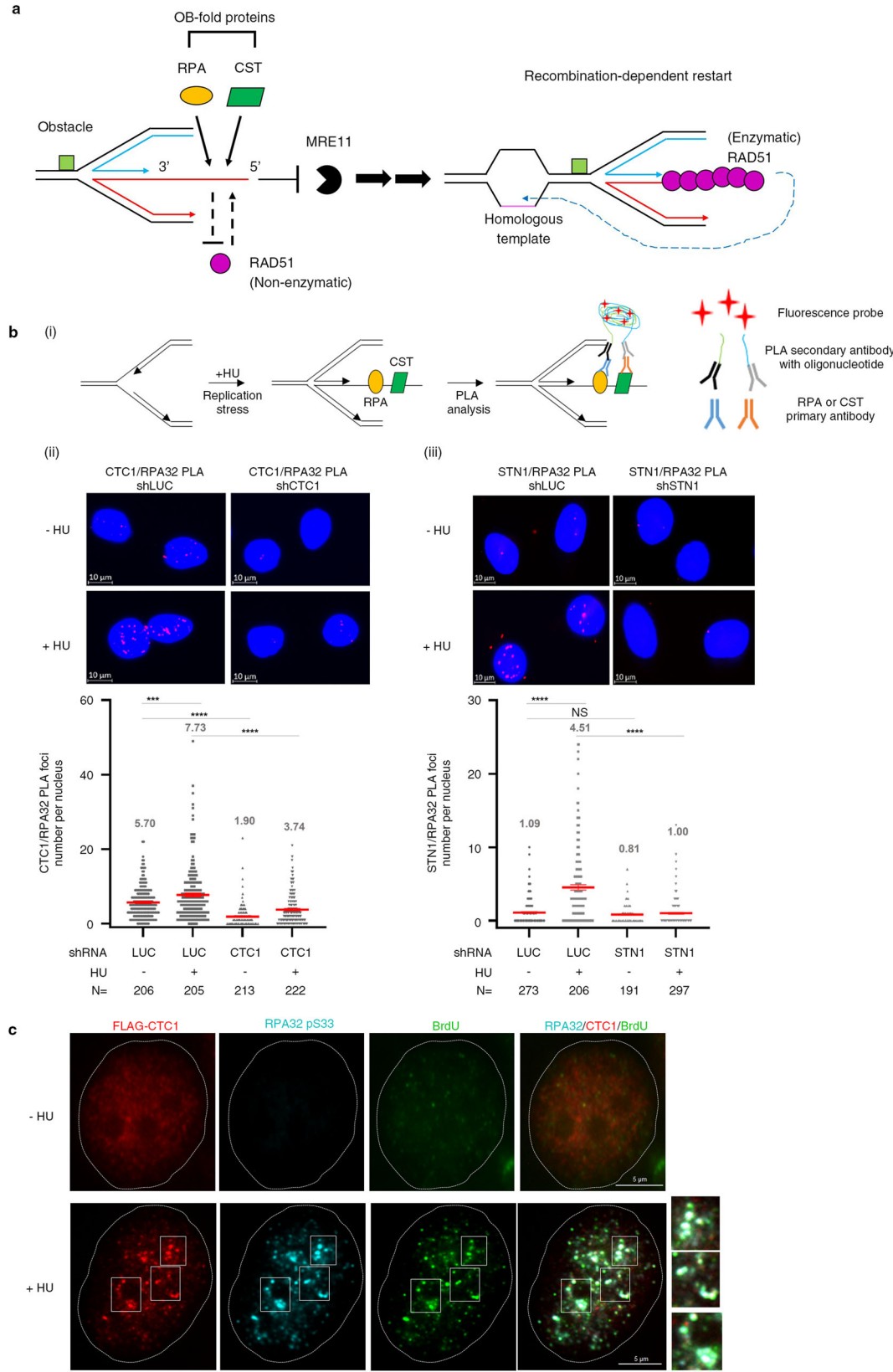

anti-CTC1/STN1 antibodies (Fig. 1c). To determine if such close proximity was due to CST and RPA colocalization at stalled forks, we overexpressed the CST complex by co-transfecting Flag-CTC1, Myc-STN1 and HA-TEN1 into HeLa cells, cultured the cells in BrdU-containing media, and then treated them with HU. Cells were then triple-stained with anti-BrdU (to detect ssDNA), anti-RPA32 pS33 (to specifically identify the 32 kDa subunit of RPA that is phosphorylated upon fork stalling), and anti-Flag (to capture CTC1 signal). Co-immunostaining revealed that in response to fork stalling, the majority of CST complex was colocalized with RPA and ssDNA (Fig. 1d). Thus, CST and RPA localize in close proximity to each other.

**Fig. 1 CST and RPA localize in close proximity on ssDNA in cells in response to replication stress. a** Crosstalk among CST, RPA, and RAD51 on ssDNA subjected to replication stress. **b** CST lies in close proximity to RPA upon fork stalling. (i) PLA is a technique that detects the physical proximity of two different proteins. In principle, if the two proteins are <40 nm apart, fluorescence signal can be detected. In brief, the two target proteins are bound by specific primary antibodies. If the target proteins are sufficiently proximal, PLA secondary antibodies hosting oligonucleotides can be ligated by means of two PLUS/MINUS PLA oligos to circularize. The DNA polymerase phi29 then processes rolling-circle amplification, and the resulting copies can be detected by hybridizing the fluorescence-labeled oligonucleotide[51]. (ii), (iii) PLA assays were performed to establish the close proximity of CTC1/STN1 with RPA in HeLa cells treated with hydroxyurea (HU) for 3 h. Representative PLA images of CTC1/RPA32 (ii) and STN1/RPA32 (iii) are shown. Scatter plots from one experiment are shown here. Red lines represent mean values ± SEM. N: the number of cells analyzed in each sample. *P* values were calculated by one-way ANOVA. NS not significant, ***$P < 0.001$; ****$P < 0.0001$. **c** CST colocalizes with RPA on the same ssDNA in response to fork stalling. HeLa cells expressing Flag-CTC/Myc-STN1/HA-TEN1 were labeled with BrdU and treated with or without HU (3 h), followed by co-immunostaining with anti-Flag (red), anti-RPA32 pS33 (cyan), and anti-BrdU (green) antibodies. $N = 3$ biologically independent experiments.

**DNA-binding properties of the CST and RPA complexes**. To elucidate the functional relationship between CST and RPA at stalled forks, we used single-molecule fluorescence resonance energy transfer (smFRET) to compare their ssDNA-binding properties[52]. Cy3- (donor dye) and Cy5-labeled (acceptor dye) substrates were employed for total internal reflection fluorescence microscopy (TIRFM) to monitor changes in smFRET intensity upon protein binding (Fig. 2a). smFRET experiments measure the change in distance between dye pairs. Upon proteins binding to DNA, the separation between dye pairs increases, thereby diminishing smFRET values. Thus, smFRET experiments enable sensitive determination of protein binding to DNA in real time. CST presented relatively weaker and more dynamic DNA-binding ability in a high-salt buffer (150 mM KCl, $Kd_{CST} = 0.29$ nM) relative to a low-salt buffer (50 mM KCl, $Kd_{CST} = 0.12$ nM; Fig. 2b and Supplementary Fig. 1a, b). Moreover, in a low-salt buffer, CST exhibited static binding and a comparable DNA-binding affinity to that of RPA ($Kd_{CST} = 0.12$ nM, $Kd_{RPA} = 0.12$ nM; Fig. 2b and Supplementary Fig. 1a, b).

Given that CST and RPA present similar DNA-binding affinities under low-salt conditions, next we examined if CST can compete with RPA for ssDNA binding by means of an ssDNA pulldown experiment (Fig. 2c, panel i). RPA in excess was preincubated with magnetic streptavidin beads linked to biotinylated 80-nucleotide (nt) ssDNA to ensure that all ssDNAs were saturated with RPA. Then we added CST to complete the reaction, before capturing the magnetic ssDNA and its associated proteins using a magnetic beads separator. We found that CST associated with the RPA-bound ssDNA in 50 mM KCl in a dosage-dependent manner and, surprisingly, that amounts of RPA on the ssDNA substrate were the same regardless of whether CST was present or not (Fig. 2c, panels ii and iii). This result indicates that CST coexists with RPA on the same ssDNA. Interestingly, co-existence of CST with RPA-bound ssDNA is abolished in a high-salt buffer (150 mM KCl; Supplementary Fig. 2a), reflecting that CST displays a similar DNA-binding affinity as RPA under conditions of lower ionic strength, as documented above. Consistent with that notion, we found that RPA could coexist with CST-bound ssDNA under low-salt (50 mM KCl) but not high-salt conditions (Supplementary Fig. 2b). Next, we examined if CST anchors the RPA-bound ssDNA via protein-protein interactions. However, no physical interaction between CST and RPA was observed in affinity pulldown assays in the absence of DNA or in the presence of 30-nt ssDNA (Fig. 2d). When we examined the physical relationship between CST and RPA by means of co-immunoprecipitation from cells, consistently we detected a physical association between CST and RPA70 (the 70 kDa subunit) that was sensitive to benzonase treatment, further supporting that CST and RPA co-occupy the same DNA molecule (Fig. 2e). Notably, *E. coli* single-stranded binding protein (SSB) lacks the ability to bind RPA-

coated ssDNA (Supplementary Fig. 2c). Together, these findings demonstrate that CST and RPA can coexist on ssDNA.

**CST and RPA occupy the same DNA molecule via facilitated dissociation**. CST and RPA harbor 9 and 6 OB-fold domains, respectively[47,53,54]. Notably, our kinetic studies and those of others have revealed that both CST and RPA exhibit concentration-dependent dissociation from ssDNA[23,55] (Supplementary Fig. 3), indicating that those protein complexes switch between free and ssDNA-bound states when free proteins are present. This property of facilitated dissociation appears to arise from the differing ssDNA-binding affinities of individual OB-fold domains[55–57]. Accordingly, the higher DNA-binding affinity of CST's OB-fold domains could outcompete the lower DNA-binding affinity of RPA's OB-fold domains for the same DNA-binding site on the same ssDNA molecule (see Fig. 3a). To explore this feature further, we used smFRET to analyze if CST occupies the same RPA-bound DNA molecule. We used Cy3- and Cy5-labeled overhang substrates to examine co-occupation of RPA-bound DNA molecules by CST and RPA (Fig. 3b, panel i). Since both CST- and RPA-bound ssDNA display smFRET values of ~0.2 in 50 mM KCl (Supplementary Fig. 1c, upper panel), making it difficult to distinguish between RPA- and CST-bound DNA, we shifted the salt condition to 150 mM KCl for a better resolution (Supplementary Fig. 1c, lower panel). RPA was pre-incubated with the DNA substrate, and then CST was added to the reaction after washing the channel. Changes in smFRET signal intensity were monitored in real time. The smFRET peak intensity of bare DNA is high (~0.65), but shifted to a low smFRET state upon inclusion of CST or RPA (~0.55 and 0.3, respectively). These distinct peak intensities allowed us to distinguish CST-bound from RPA-bound ssDNA (Fig. 3b, panels i–iii). Surprisingly, we found that the RPA signal shifted to an intermediate position between the RPA and CST smFRET peaks (~0.4) when we added CST into the reaction, indicating that CST may occupy the RPA-bound DNA molecule (Fig. 3b, panels iv–v).

In parallel with smFRET, we also employed single-molecule fluorescence colocalization single-molecule spectroscopy (CoSMoS)[58] to confirm that both CST and RPA occupy the same DNA molecule. Fluorescent EGFP-tagged RPA and SNAP-tagged CST labeled with SNAP-surface 649 fluorescent dye were purified for this purpose. Both of these tagged proteins behaved like native proteins, as evidenced by their DNA-binding affinity profiles (Supplementary Fig. 4). EGFP-RPA was incubated on a slide containing Cy3-labeled, 80-nt ssDNA overhang substrates. After washing away free RPA, we added the SNAP$_{649}$-labeled CST (Fig. 3c, panel i). Green, blue, and red fluorescence signals represent ssDNA, RPA, and CST, respectively (Fig. 3c, panels ii–iv). Colocalization of EGFP and Cy3 fluorescence identified RPA-bound DNA, revealing that ~85% of DNA foci are RPA-bound (Fig. 3c, panels ii, iii, and vi). As shown in panels iii and iv

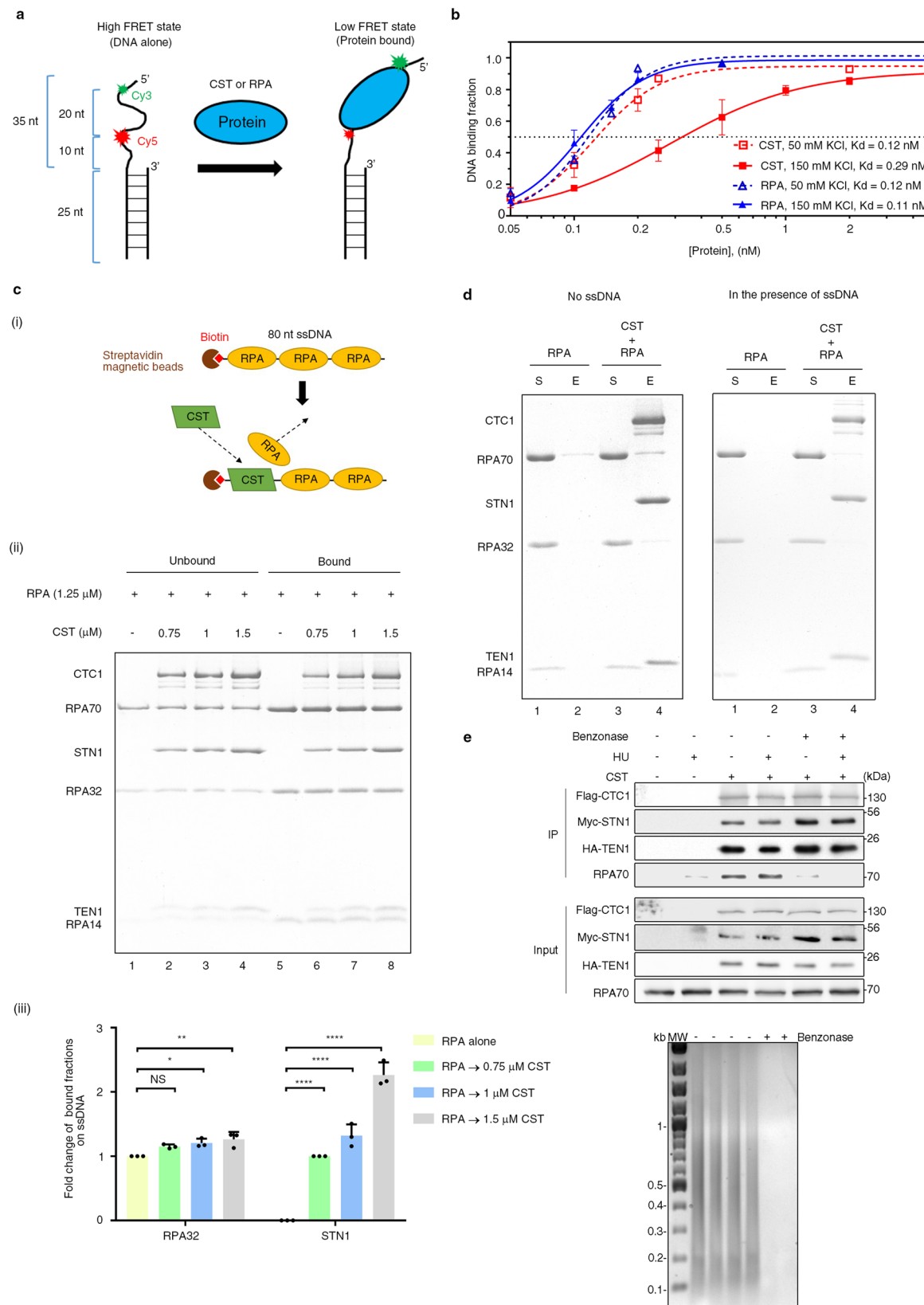

of Fig. 3c, some of the RPA-DNA foci colocalized with CST, indicative of CST and RPA co-occupancy on the same DNA molecule. The percentage colocalization of CST with RPA-coated ssDNA is CST concentration-dependent, whereas that of RPA with ssDNA is not (Fig. 3c, panels v and vi).

Since the DNA-binding size of RPA is about 20–30 nt, it is possible that up to four RPA molecules are bound in our 80-nt ssDNA substrates[3,53,54]. We investigated if some of the RPA molecules are displaced upon CST-RPA-DNA colocalization. As individual fluorescent dyes display stochastic photobleaching, we

**Fig. 2 DNA-binding characteristics of the CST and RPA complexes. a** Schematic showing the design of our single-molecule FRET (smFRET) experiment to determine DNA-binding affinities. **b** Measurement of the DNA-binding affinity (Kd) of CST and RPA in ionic strengths of 50 mM and 150 mM KCl. The curve was fitted by means of a Hill slope equation in GraphPad Prism. At 50 mM KCl, the Kd values of RPA and CST are 0.12 nM (Hill slope = 3.14) and 0.12 nM (Hill slope = 2.74), respectively. At 150 mM KCl, the Kd values of RPA and CST are 0.11 nM (Hill slope = 2.83) and 0.29 nM (Hill slope = 1.4), respectively. Data points of each protein concentration represent mean ± S.D. calculated from three independent experiments. **c** (i) Illustration of our ssDNA pulldown assay. Excessive RPA was preincubated with a biotinylated 80-nt ssDNA linked to magnetic streptavidin beads. After adding CST, the ssDNA and its associated proteins were captured using a magnetic bead separator. (ii) RPA was preincubated with magnetic ssDNA beads and then the indicated amounts of CST were added under the condition of 50 mM KCl. The unbound and bound fractions from the reaction were analyzed by 15% SDS-PAGE with Coomassie blue staining. (iii) Quantitative plot of amounts of RPA32 and STN1 in the bound fraction. Data represent mean ± S.D. calculated from three independent experiments. NS, not significant, *$P < 0.05$; **$P < 0.01$; ***$P < 0.001$; ****$P < 0.0001$. Statistical analyses were performed by one-way ANOVA with Tukey's post hoc test. **d** Affintiy pulldown assay. Flag-CTC1-STN1-TEN1-His$_6$ (1 μM) was incubated with RPA (1 μM) in the absence of DNA or in the presence of 30-nt ssDNA, followed by incubation with His-Tag Dynabeads to capture CST and associated proteins using a magnetic bead separator. The supernatant (S) and eluate (E) were analyzed. RPA alone is shown as a control. $N = 3$ biologically independent experiments. **e** CST physically associates with RPA in a DNA-dependent manner. FLAG-CTC1, Myc-STN1, and HA-TEN1 were co-expressed in HEK293T cells and then treated with 2 mM HU for 20 h. Cell lysates were treated with or without benzonase prior to immunoprecipitation with anti-Myc. Top: Western blot. Bottom: Agarose gel analysis of DNA removal after benzonase treatment. $N = 3$ biologically independent experiments.

used a single-molecule fluorescence photobleaching experiment to quantify the number of dye-labeled RPA molecules on individual DNA molecules. In Fig. 3d, we show a representative time-course of a four-step photobleaching experiment on EGFP-RPA. We compared the amounts of RPA molecules on 80-nt ssDNA substrates in the presence or absence of CST molecules (Fig. 3e), which revealed a distribution of 1–4 bound RPA molecules and this distribution was the same with or without CST, indicating that RPA molecules had not been displaced upon binding of CST. These analyses further validate that CST and RPA can coexist on the same ssDNA molecule.

**CST can recruit RAD51 onto RPA-bound ssDNA.** Next, we examined how RPA and CST influence the assembly of RAD51 filaments. It has been well documented that RPA suppresses the assembly of RAD51 filaments owing to RPA's higher affinity for ssDNA[23]. Interestingly, our recent studies have shown that CST immunoprecipitates with RAD51 and facilitates the recruitment of RAD51 at stalled forks in response to HU treatment[31,32]. Given that CST can compete with RPA for ssDNA binding, as shown in the current work, we hypothesized that CST could target RAD51 to RPA-bound ssDNA. To test that possibility, we examined the interaction between purified CST and RAD51 proteins by affinity pulldown. That experiment showed that CST physically interacts with RAD51 but not with the *E. coli* homolog RecA (Fig. 4a and Supplementary Fig. 5a). Next, we performed a ssDNA pulldown assay to establish if CST could target RAD51 to the RPA-bound ssDNA. In brief, RPA was preincubated with magnetic ssDNA beads and then CST and/or RAD51 were added to the reaction mixture (Fig. 4b, panel i). As anticipated, RAD51 alone did not bind to RPA-bound ssDNA (Fig. 4b, panel ii, compare lanes 3 and 7). Importantly, RAD51 was captured by the RPA-bound ssDNA under the same conditions, but with the added presence of CST (Fig. 4b, panel ii, compare lanes 7 and 8, and panel iii). Notably, RecA cannot be captured by the RPA-bound ssDNA, even in the presence of CST (Supplementary Fig. 5b). We also explored if the DNA-binding ability of CST is a prerequisite for the recruitment of RAD51 to RPA-bound ssDNA. In previous studies[31,46], we demonstrated that removal of the N-terminal 700 residues of CTC1 (CTC1Δ700N) does not affect complex formation with STN1 and TEN1, but the mutant CTC1Δ700N-ST complex lacks DNA-binding activity. Here, although we found that CTC1Δ700N-ST can still interact proficiently with RAD51, albeit to a lesser extent than for wild-type CST (Supplementary Fig. 5c), it fails to recruit RAD51 to RPA-bound ssDNA (Fig. 4c). Thus, CST appears to target RAD51 to RPA-bound ssDNA via its DNA-binding activity.

**RAD51 catalyzes DNA strand exchange of CST-bound ssDNA.** Our above-documented single-molecule analyses have revealed the salt-dependent dynamic DNA-binding properties of CST. We wanted to examine if CST suppresses RAD51-mediated DNA strand exchange under different salt conditions. We preincubated RPA/CST with ssDNA before adding RAD51 and then initiated a reaction by adding radiolabeled homologous duplex DNA (Fig. 5a). We found that RPA strongly inhibited RAD51-mediated strand exchange under our different salt conditions (Fig. 5b). Interestingly, CST did not inhibit RAD51 activity at 100 or 150 mM KCl (Fig. 5c). Indeed, RAD51 activity was not suppressed even though amounts of CST were 2–4-fold higher than for RPA (Fig. 5d). Note that calcium was included in the reaction to enhance RAD51 activity. We also obtained the same outcome when we replaced calcium and ATP with magnesium and non-hydrolyzable AMPPNP to boost RAD51 activity (Fig. 5e).

**Assembly of RAD51 filaments on CST-bound ssDNA.** Next, we examined if RAD51 can form nucleoprotein filament on CST-bound ssDNA in 150 mM KCl. To do so, we employed electron microscopy (EM) with negative staining to image the formation of RAD51 filaments on either CST-ssDNA or RPA-ssDNA substrate. As expected, RAD51 formed helical filaments (median length 96.41 nm) on naked 80-nt ssDNA whereas, as anticipated, no filament structures were observed for CST-ssDNA or RPA-ssDNA substrate alone (Fig. 6a, panels i–iii). Most importantly, whereas RAD51 only formed fragmented and short filaments (median length 67.83 nm) on RPA-bound ssDNA, we observed longer RAD51 filaments (median length 99.17 nm) on CST-bound ssDNA (Fig. 6a, compare panels iv and v, see quantification in Fig. 6b).

In parallel with our EM analysis, we performed smFRET to study the dynamics and kinetics of RAD51 filament assembly on CST- or RPA-bound ssDNA substrate (described above and in Fig. 2a) in 150 mM KCl. The smFRET peak intensities of CST-, RPA-, and RAD51-ssDNA filaments are ~0.2, 0.15, and 0.05, respectively (Fig. 6c, open bars). RPA-ssDNA peak intensity remained unchanged upon introducing RAD51, implying that no RAD51 filament had formed (Fig. 6c, panel iii). In contrast, adding RAD51 to CST-coated-ssDNA shifted some of the smFRET peaks from 0.2 to 0.05, indicating that some of the CST-ssDNA can form RAD51 filament (Fig. 6c, panel v). We also examined the assembly of RAD51 filaments on CST-coated ssDNA in real time by introducing RAD51 into the reaction chamber containing preformed CST-ssDNA complex (Fig. 6d, gray area = dead-time). In the representative time-course shown in Fig. 6d, the CST-ssDNA state persists over 25–100 s (blue

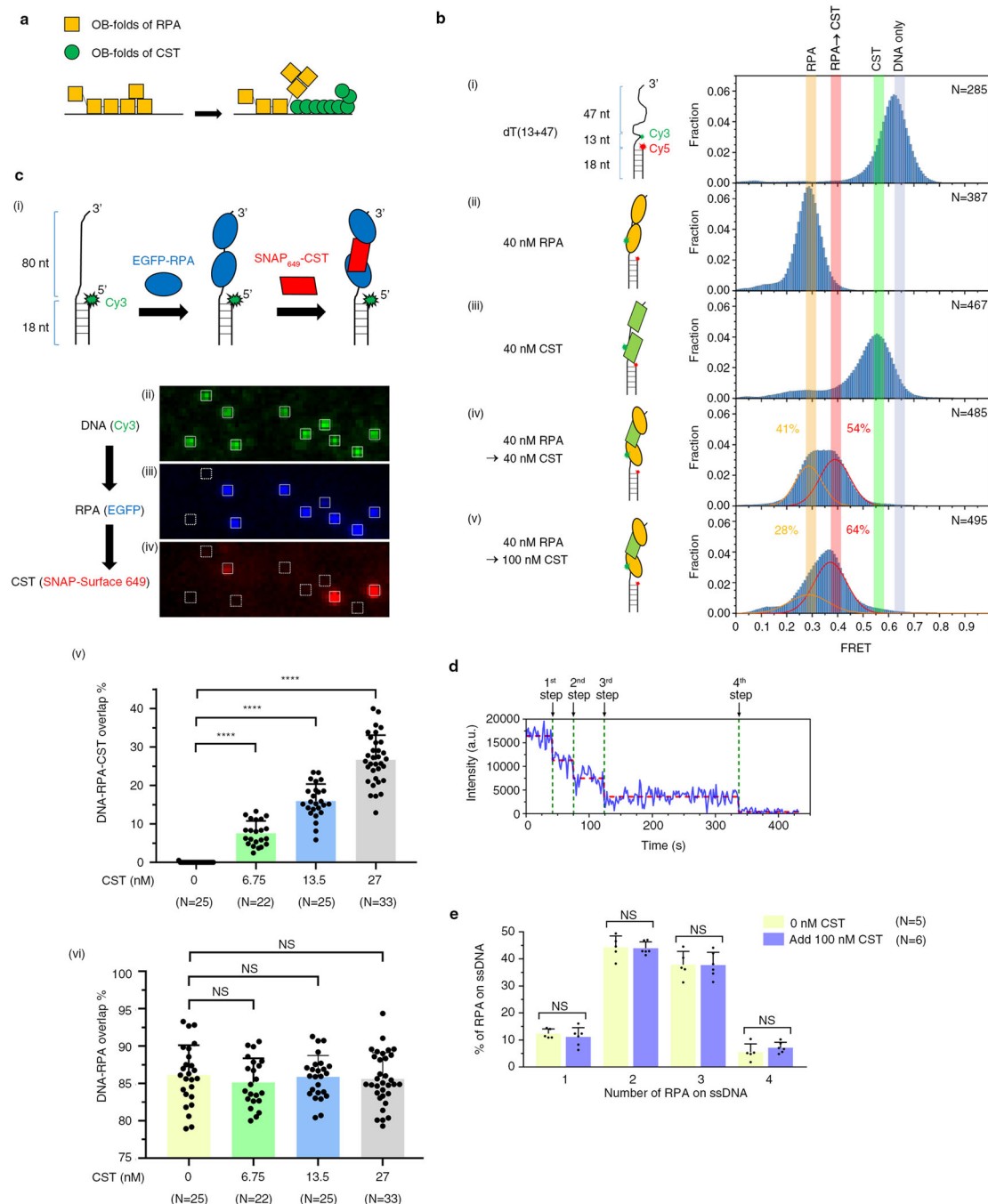

area), followed by a transition to a high-FRET intermediate state (green area) at 115 s. At ~120 s, the smFRET peak intensity transitions to ~0.05, reflective of the RAD51-bound state (pink area). RAD51 nucleoprotein assembly on CST-coated ssDNA requires the transition to the high-FRET state (green area). This transition to the high-FRET state is most apparent for the case of RAD51 assembly, as evident in the compiled rastergram presented in Fig. 6d. As FRET experiments describe separation between the dye pairs, the exact nature of this high-FRET state remains to be characterized. It is possible that this high-FRET intermediate state could arise from partial dissociation of CST or transient interaction of CST or the CST-RAD51 complex. Therefore, the dynamic nature of CST likely contributes to and facilitates the RAD51 assembly observed for CST-coated ssDNA but not RPA-coated ssDNA. Our kinetic analysis shows that it takes about 60 s for a RAD51 filament to assemble on CST-coated

ssDNA, whereas <10% of the RPA-coated ssDNA was amenable to RAD51 nucleoprotein filament assembly (Fig. 6e). Thus, our EM and smFRET analyses indicate that RAD51 can form nucleoprotein filaments on CST-coated ssDNA.

Finally, we wondered if CST functions as a mediator to facilitate utilization of RPA-coated ssDNA as a strand-exchange substrate. In a DNA strand-exchange assay, we preincubated RPA with ssDNA and then monitored RAD51 activity in the presence or absence of CST (Supplementary Fig. 6a). However, we found that CST could not overcome the inhibitory effect of RPA on RAD51-mediated DNA strand exchange (Supplementary Fig. 6b, c). In parallel with the DNA strand-exchange assay, we conducted a D-loop formation assay to examine CST mediator activity. Our results show that inclusion of CST in the reaction did not promote RAD51-mediated homologous DNA pairing and strand exchange activity with the RPA-coated ssDNA substrate

**Fig. 3 Single-molecule experiments show that CST coexists with RPA on ssDNA. a** Illustration showing that CST and RPA can co-occupy the same ssDNA via competion between the OB-fold domains of different molecules. The high ssDNA-binding affinity of CST's OB-fold domains can compete with the RPA's low-affinity OB-fold domains for the same DNA-binding site. **b** smFRET reveals that CST coexists with RPA by using a dT(13 + 47) DNA overhang substrate. smFRET histograms were generated from numerous smFRET measurements of individual molecules. N values represent the number of individual molecules collected from three independent experiments and are displayed in the upper right corner. (i) For DNA alone, the smFRET intensity was ~0.65. (ii) When DNA was incubated with 40 nM RPA, the smFRET intensity was ~0.3. (iii) When DNA was incubated with 40 nM CST, the smFRET intensity was ~0.55. (iv and v) When DNA was preincubated with 40 nM RPA and then incubated with 40 nM or 100 nM CST after washing out RPA, the smFRET intensity shifted from ~0.3 to an intermediate value of ~0.4. The percentage of RPA-ssDNA and the intermediate state are shown on the graph. **c** Colocalization single-molecule microscopy (CoSMoS) reveals that CST coexists with RPA on the same DNA molecule. (i) Schematic of our CoSMoS assay. Approximately 15 pM Cy3-labeled 80-nt overhang DNA substrate (green) was tethered on the slide surface. Then, 10 nM EGFP-RPA (blue) was incubated with the DNA substrate at 50 mM KCl, before incubating with 13.5 nM SNAP$_{649}$-CST (red) after RPA washout. (ii–iv) The fluorescence images acquired from three different emission band-pass filters in the same surface region at the three reaction stages corresponding to (i). (v) Percentage colocalization of DNA, RPA, and CST for the indicated amounts of added CST. (vi) Percentage colocalization of DNA and 10 nM RPA for the indicated amounts of added CST. Data represent mean ± S.D. N values represent the microscope fields of view that were assessed from three independent experiments. NS not significant, ****$P < 0.0001$, as calculated by one-way ANOVA with Tukey's post hoc test. **d** Single-molecule fluorescence photobleaching experiment showing four-step photobleaching, indicating that there are four EGFP-RPA molecules on the DNA substrate. **e** Distribution of amounts of RPA molecules bound on ssDNA substrates based on the photobleaching experiments. There is no statistical difference between presence or absence of 100 nM CST. Data represent mean ± S.D. N values represent the number of experiments. NS, not significant, as calculated by Student's $t$ test (two-tailed) with correction for multiple comparision using the Holm–Sidak method.

---

(Supplementary Fig. 6d, e). In conclusion, our data indicate that although CST can recruit RAD51 to RPA-bound ssDNA, it lacks recombination mediator activity.

**Discussion**

Both RPA and CST harbor multiple OB-fold domains and form a heterotrimeric complex. Although both complexes exhibit a high affinity for ssDNA, the CST complex presents a differential response to ssDNA depending on the length and sequence of this latter. For instance, CST has a high affinity for an 18-nt ssDNA with a G-rich sequence, but sequence specificity is diminished if the oligonucleotide is longer[46,59]. Interestingly, our smFRET experiments demonstrate that ionic strength significantly alters the DNA-binding dynamics of the CST complex. We found that under high-salt (150 mM KCl) conditions, CST exhibits dynamic association/dissociation with ssDNA, whereas RPA remains relatively static on ssDNA (Supplementary Fig 1). Their unique ssDNA-binding characteristics highlight the complexity and flexibility underlying regulation of RPA and CST, which enables these two ssDNA-binding proteins to fulfill their physiological roles.

ChIP analyses have shown that CST deficiency significantly reduces RAD51 recruitment to telomeric and non-telomeric GC-rich sequences under replication stress[32]. Single-strand G-rich repetitive sequences are prone to forming G-quadruplex structures, indicating a potential function for CST-RAD51 in G4 secondary structures. Recent cellular and single-molecule studies have documented that CST can unfold G4 structure and is recruited to telomeric and non-telomeric DNA in response to G4 formation[55,60]. Furthermore, CST prevents G4-induced inhibition of DNA replication, and CST depletion slows lagging strand telomere replication after G4 stabilization[60]. These data indicate that CST unfolds the G4 secondary structure to prevent or resolve replication stalling at G4 sites. Therefore, we hypothesize that CST may prevent the accumulation of G4 secondary structures to allow efficient binding of RAD51 to ssDNA for replication fork restart.

As a core DNA replication factor, RPA is significantly more abundant than CST in cells[61,62]. Notably, CST is not only detected at telomeres but is also recruited to fragile sites under replication stress, supporting a specific role for CST in response to replication stress[32,63]. Consistent with previous observations[32,48,49], we show that CST and RPA colocalize at stalled replication forks (Fig. 1). This finding raises the intriguing question as to how RPA and CST coordinate with each other to engage with the ssDNA formed at stalled forks. Our biochemical and single-molecule data presented herein demonstrate that CST harbors a similar affinity for ssDNA as RPA under low-salt conditions (50 mM KCl). Most importantly, our ssDNA pull-down assay and smFRET analyses provide evidence that CST and RPA can co-occupy the same ssDNA molecule. Two possible binding patterns can be deduced from their ssDNA co-occupancy (Supplementary Fig. 7a, b). One is that CST and RPA bind to separate regions of the same ssDNA molecule (Supplementary Fig. 7a). Alternatively, if the length of ssDNA is limited, the different ssDNA-binding affinities of each OB-fold domain in the CST or RPA complexes could contribute to their ability to co-occupy ssDNA (Supplementary Fig. 7b). By directly observing the localization of fluorescing CST and RPA, our single-molecule CoSMoS analysis suggests that both CST and RPA protein complexes can colocalize on the same ssDNA (Fig. 3c). Consistent with this notion, it is clear that both CST and RPA possess facilitated dissociation characteristics (Supplementary Fig. 3), indicating that those proteins undergo rapid exchange between bound and unbound states when free proteins are present[55–57]. Taken together, these results suggest that CST OB-fold domains with the highest ssDNA-binding affinity can compete with RPA OB-fold domains with lower ssDNA-binding affinity via a facilitated exchange mechanism. Accordingly, there is no complete dissociation of either protein complex from the ssDNA. How the OB-fold domains of CST and RPA compete for DNA sites needs to be further explored by isolating OB-fold point mutants of RPA and CST.

RAD51 plays an important role in the remodeling and restart of stalled/collapsed replication forks under replication stress. Notably, depletion of CST attenuates the recruitment of RAD51 to stalled replication forks[31,32], implying a functional interaction between CST and RAD51. Consistent with that notion, RAD51 has been shown to be immunoprecipitated by CST and both proteins colocalize in response to HU treatment[31,32]. Moreover, our in vitro affinity pulldown assay further revealed that CST directly interacts with RAD51 (this study, Fig. 4a). To address the functional significance of CST and RAD51 interaction, previously we generated clinically relevant missense and small-deletion pathogenic CTC1 mutant variants to characterize their functional interaction with RAD51[49]. Among them, the CTC1 R975G and CTC1 C985Δ mutants retain CST complex formation ability, but impair the interaction with RAD51[49]. Although both variants

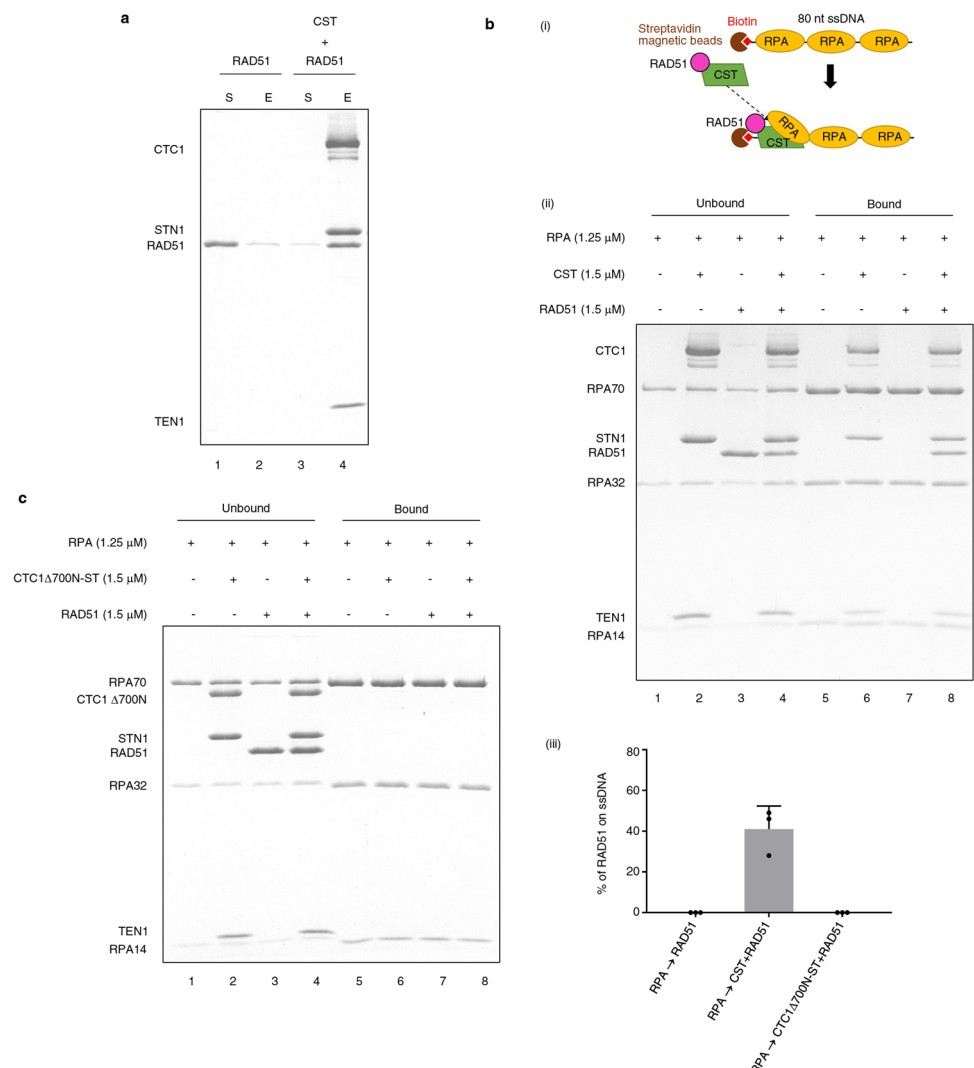

**Fig. 4 CST physically interacts with RAD51 and targets it to RPA-coated ssDNA. a** Affinity pulldown assay. Flag-CTC1-STN1-TEN1-His$_6$ (1 μM) was incubated with RAD51 (1 μM), followed by incubation with His-Tag Dynabeads to capture the CST and associated proteins by means of a magnetic bead separator. The supernatant (S) and eluate (E) were analyzed by 15% SDS-PAGE with Coomassie blue staining. RAD51 alone is shown as a control. $N = 3$ biologically independent experiments. **b** (i) Schematic of the ssDNA pulldown experiment. (ii) Excessive RPA was preincubated with biotinylated 80-nt ssDNA linked to magnetic streptavidin beads. Upon addition of CST and RAD51, the ssDNA and its associated proteins were captured using a magnetic bead separator. The unbound and bound fractions from the reaction were analyzed by 15% SDS-PAGE with Coomassie blue staining. (iii) Quantitative plot of amounts of RAD51 in the bound fraction. Data represent mean ± S.D. calculated from three independent experiments. **c** For ssDNA pulldown analysis, RPA was preincubated with magnetic ssDNA beads. Then CTC1Δ700N-ST and RAD51 were added to complete the reaction. The unbound and bound fractions from the reaction were analyzed by 15% SDS-PAGE with Coomassie blue staining. $N = 3$ biologically independent experiments.

exhibit global genome instabilities, which are further elevated by replication stress, purified CTC1 R975G and CTC1 C985Δ proteins also lack DNA-binding activity[34]. Thus, it will be important in the future to identify distinct separation-of-function CST variants that are specifically defective in the RAD51 interaction to address this issue further. Since CST directly interacts with RAD51, not RecA, and does not recruit RecA to RPA-bound ssDNA (this work, Supplementary Fig. 5a, b), our results support the notion that the physical interaction between CST and RAD51 is at least partially responsible for RAD51 recruitment to RPA-bound ssDNA. Apart from the physical interaction, recruitment of RAD51 requires the DNA-binding ability of CST since the CTC1Δ700N-ST mutant that is defective in DNA-binding but largely retains RAD51 interaction ability cannot target RAD51 to RPA-coated ssDNA. In support of this scenario, a previous cell-based study showed that loss of CST's DNA-binding ability significantly attenuated HU-induced RAD51 foci formation[31]. These

outcomes emphasize that the DNA-binding activity of CST is a prerequisite for efficient RAD51 recruitment to stalled replication forks.

Single-stranded DNA represents an intermediate configuration in many aspects of DNA replication and repair. Accordingly, how the ssDNA-binding affinity of CST and RPA is regulated influences the assembly of RAD51 filaments. For example, during HR, RPA-coated ssDNA prevents RAD51 filament assembly, with the tumor suppressor BRCA2 acting to overcome that inhibitory activity[25]. Although CST can target RAD51 onto RPA-coated ssDNA, we did not detect any RAD51-driven strand-exchange products under our low-salt conditions. This outcome is consistent with the notion that RPA remains on ssDNA and that RAD51 is tethered by CST via an RPA-CST-facilitated dissociation mechanism on the same ssDNA molecule, as described above. It is tempting to hypothesize that the dynamic ssDNA-binding property of CST facilitates recruitment of RAD51 to

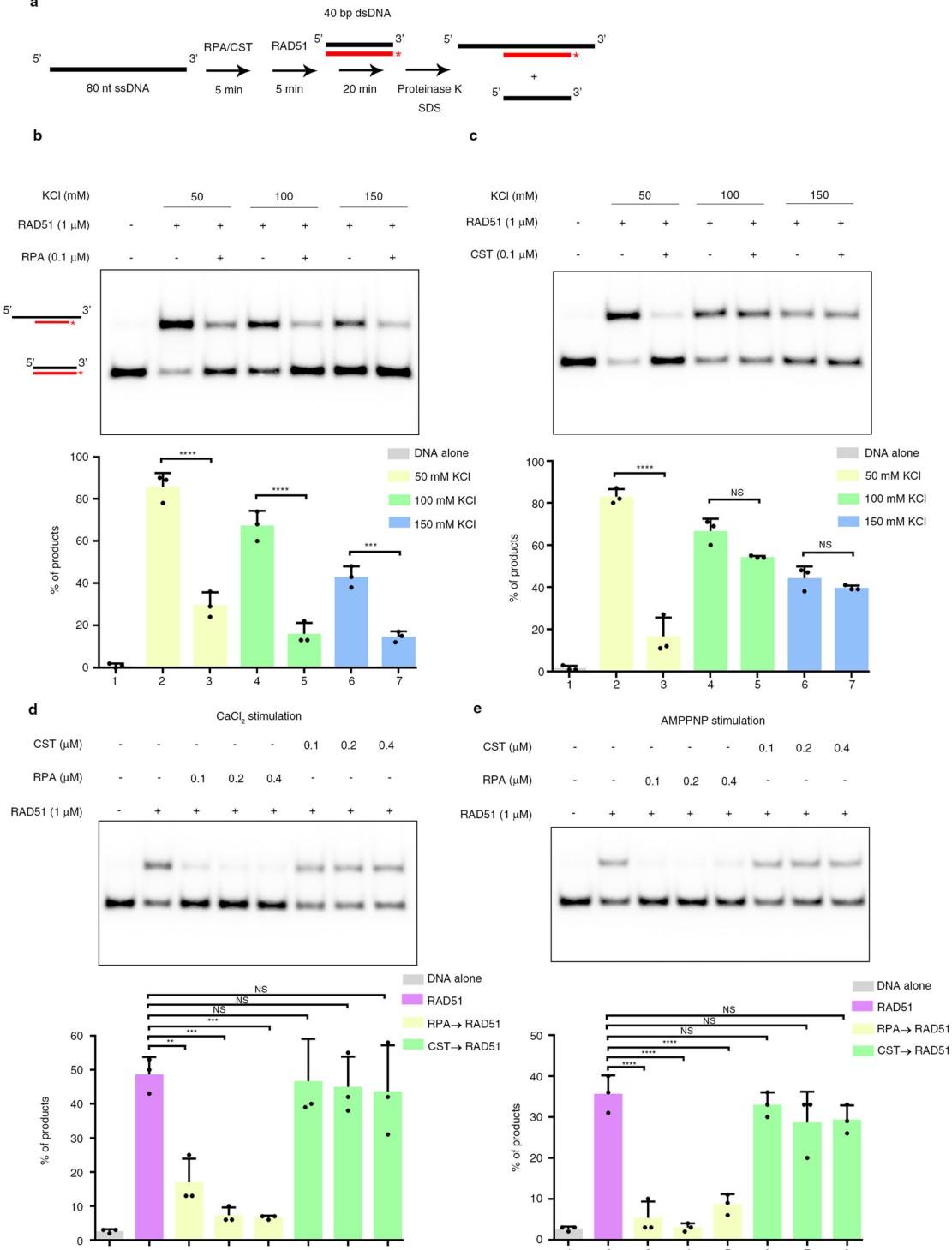

**Fig. 5 The effects of RPA and CST on RAD51-mediated strand exchange. a** Schematic of the DNA strand-exchange assay. The $^{32}$P-labeled DNA is marked by an asterisk. **b, c** RPA (**b**) or CST (**c**) was preincubated with ssDNA at the indicated concentrations of KCl. RAD51 activity was determined by monitoring the formation of strand-exchange products. Note that 5 mM CaCl$_2$ was included in the reaction to stimulate RAD51 activity. **d, e** The indicated amounts of RPA and CST were preincubated with ssDNA and 150 mM KCl before adding RAD51. CaCl$_2$ (**d**) or AMPPNP (**e**) was used to stimulate RAD51 activity. **b–e** Quantitative plots are shown below the gel images. Data represent mean ± S.D. calculated from three independent experiments. NS not significant, **$P < 0.01$; ***$P < 0.001$; ****$P < 0.0001$, as calculated by one-way ANOVA with Tukey's post hoc test.

RPA-coated ssDNA at stalled forks and assists in the displacement of RPA by BRCA2 (see model, Supplementary Fig. 7c). Consistent with this idea, our recent cell-based study also documented that CST depletion elevates genome instability in BRCA2-deficient cells, based on various assays including

detection of micronuclei and anaphase bridge, γH2AX foci, BrdU incorporation signal, and chromosome aberrations[31], suggesting an additive effect of CST and BRCA2 in response to replication stress. Finally, although CST-mediated targeting of RAD51 to RPA-coated ssDNA does not result in assembly of competent

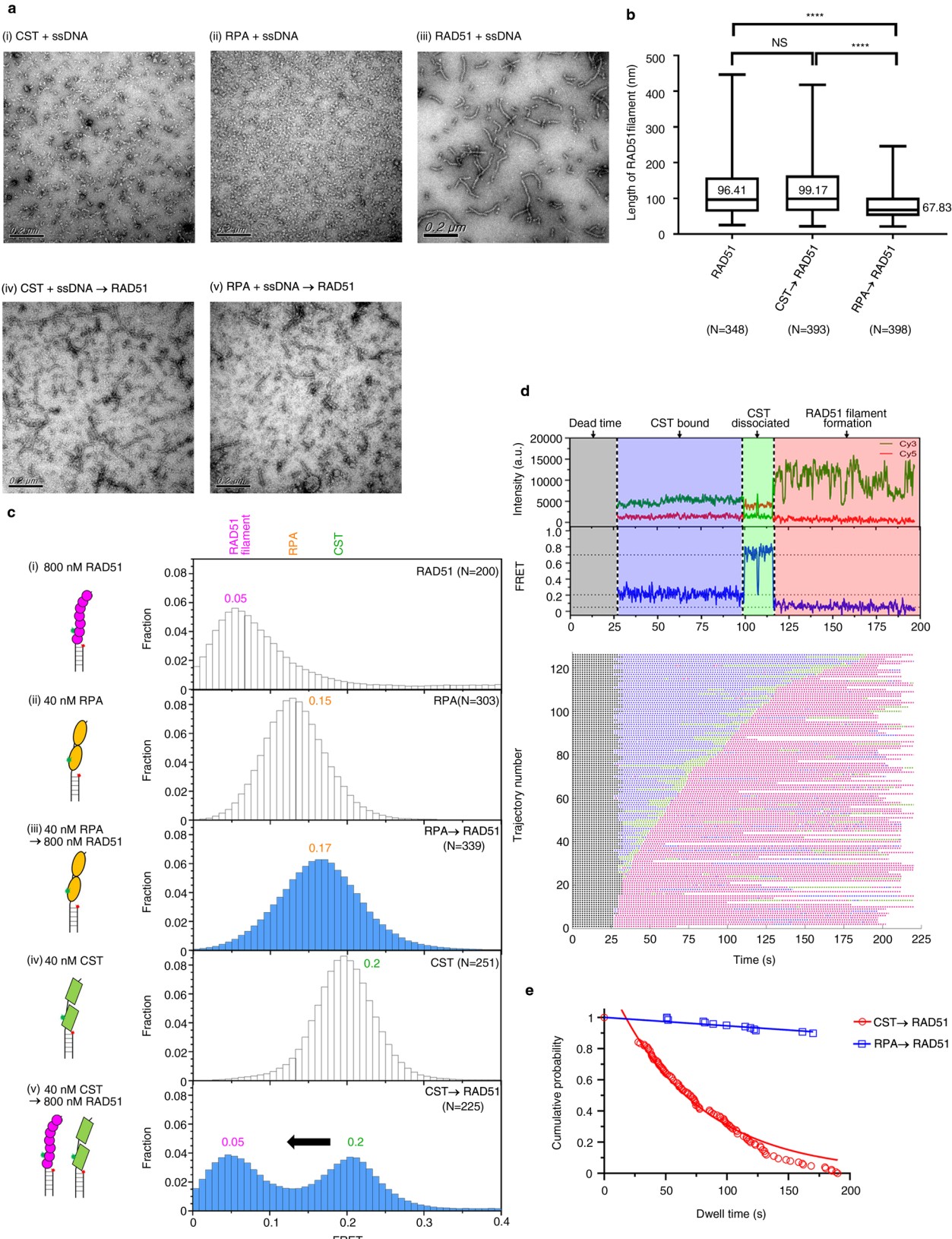

RAD51 filaments for homologous DNA pairing and strand exchange, it is possible that CST-mediated RAD51 recruitment could lead to nascent strand protection in response to replication stress. Whether the CST-RAD51 axis alone is sufficient for protection or BRCA2 activity is also a prerequisite needs to be further elucidated.

In conclusion, our study has revealed unique DNA-binding characteristics of CST. Importantly, CST but not RPA physically interacts with RAD51. We have further shown that CST recruits and tethers RAD51 to RPA-coated ssDNA via a facilitated dissociation mechanism. Our mechanistic analyses indicate that assembly of RAD51 filament during restart of stalled/collapsed

**Fig. 6 Formation of RAD51 filaments on CST-coated ssDNA. a** Electron microscopy with negative staining was used to observe the formation of RAD51 filaments on CST- or RPA-bound ssDNA under the condition of 150 mM KCl. The 80-nt ssDNA substrate was incubated with CST (i), RPA (ii) or, as a control, RAD51 (iii). RAD51 was then added to CST-bound ssDNA (iv) or RPA-bound ssDNA (v). Representative images from panels (i) to (v) are shown. $N = 2$ biologically independent experiments. **b** RAD51 filament length was measured in Image J software. The quantitative graph was generated in GraphPad Prism. Boxes represent the 25th and 75th percentiles and encompass the median, with whiskers showing the minimum and maximum values. $N$ values represent the number of RAD51 filaments identified in numerous electron microscopy images collected from two independent experiments. NS not significant, ****$P < 0.0001$ as calculated by one-way ANOVA with Kruskal–Wallis test followed by Dunn's post hoc test. The median values for RAD51, CST→RAD51, RPA→RAD51 are 96.41, 99.17, and 67.83 nm, respectively. Note that some filaments are much longer than expected due to end-to-end association of two filaments[69]. **c** RAD51 assembly on CST- or RPA-bound ssDNA was monitored by smFRET under the condition of 150 mM KCl. A 35-nt overhang DNA substrate was used for smFRET, as shown in Fig. 2a. The smFRET states of RAD51-, RPA-, and CST-ssDNA are ~0.05 (i), ~0.15 (ii), and ~0.2 (iv), respectively. Although addition of RAD51 did not alter the smFRET state of RPA-bound ssDNA (~0.15, iii), the smFRET state shifted from 0.2 to 0.05 for CST-bound ssDNA (v). This outcome indicates that RAD51 can access CST-bound ssDNA. N values represent the number of individual molecules collected from at least three independent experiments and are displayed in the upper right corner. **d** Representative time-course of RAD51 assembly on CST-coated ssDNA. RAD51 was introduced to the preformed surface-bound CST-ssDNA (gray). Before transitioning into the RAD51-bound smFRET state (pink), a high-FRET intermediate state (green) was observed. Rastergram of 126 molecules showing a similar pattern. **e** Kinetics analysis showing that RAD51 assembled on CST-coated ssDNA in $64 \pm 2.2$ s ($N = 96$), whereas it was predicted to take thousands of minutes to do so on RPA-coated ssDNA ($N = 13$). Nearly no RAD51 assembly was observed for RPA-coated ssDNA. Data are plotted as a best-fit result ± 95% C.I.

replication forks could be tightly regulated by the crosstalk between CST and RPA. Notably, ionic strength has a significant impact on the affinity and dynamics of ssDNA-binding by CST in vitro. The impact of ionic strength could be functionally replaced by other cellular factors and/or protein modifications in physiological contexts. Our findings may help delineate the disease mechanisms underlying CST mutation-related dyskeratosis and Coats plus syndrome.

## Methods

**Cell lines and cell culture**. HeLa and HEK293T cells were obtained from American Type Culture Collection (VA, USA). Cells were cultured at 37 °C and 5% $CO_2$ in DMEM media supplemented with 10% cosmic calf serum (ThermoFisher). HeLa cells depleted of CTC1 or STN1 were as described previously[32].

**Plasmids**. The expression plasmids of human CST complex and the CTC1Δ700N-ST variant were constructed as previously described[31]. The amino-terminal Flag-SNAP-tagged CTC1 expression plasmid was generated by cloning the SNAP tag into pEAK8 Flag-CTC1 plasmid. Human RAD51, RPA, and EGFP-RPA expression plasmids were constructed as previously described[23,64,65].

**Expression and purification of recombinant proteins**. Human CST complex and CTC1Δ700N-ST were expressed in Expi293F cells and purified as described previously[31]. The SNAP-CST complex was purified using the same procedures as for the wild-type complex. In brief, clarified cell extract from SNAP-CST-overexpressing Expi293F cells was subjected to Ni²⁺ NTA-agarose affinity and then anti-Flag M2 affinity purifications, before undergoing Superdex 200 Increase 10/300 GL-based fractionation[31]. SNAP-CST was thus purified to ≥95% homogeneity. Human RAD51, RPA, and EGFP-RPA were expressed in E. coli and purified as described previously[23,64,65]. SDS-PAGE gel images of the purified proteins are presented in Supplementary Fig. 4a, b. E. coli RecA and His-tagged RecA was purchased from New England Biolabs and E. coli SSB was purchased from Promega.

**DNA substrates**. All DNA sequences are shown in Supplementary Table 1. For the strand exchange assay and EM, we used 80-mer ssDNA Oligo 1 for presynaptic filament assembly. To prepare the homologous duplex 40-mer dsDNA, Oligo 2 was 5' end-labeled with [γ-³²P] ATP (PerkinElmer) and treated with T4 polynucleotide kinase (New England Biolabs). Upon removal of the unincorporated nucleotide via a Micro Bio-Spin P-6 column (Bio-Rad), the radiolabeled oligonucleotide was then annealed to its complementary sequence (Oligo 3) by heating the mixture of these two oligonucleotides at 85 °C for 3 min and then slowly cooling it from 65 to 25 °C. The resulting duplex DNA was then purified from a 10% polyacrylamide gel and concentrated in TE buffer (10 mM Tris-HCl, pH 8.0, 0.5 mM EDTA).

To prepare ssDNA streptavidin-magnetic beads for the DNA pulldown assay, 5′-end biotin-conjugated 80-mer ssDNA Oligo 1 was used to bind streptavidin-magnetic particles (Roche) as described previously[64].

For smFRET experiments, the fluorescing DNA overhang substrates were generated by using the paired oligos described in Supplementary Table 1 and gradient PCR annealing from 85 °C to 25 °C in T + 50 buffer (20 mM Tris pH 8.0, 50 mM NaCl). These hybrid DNA substrates were then anchored to a slide surface-coated with biotin-mPEG and streptavidin[66].

**smFRET experiments and data analysis**. smFRET experiments were conducted as described previously[66]. In brief, we modified PEG and biotin-PEG molecules on glass slides. Then the slides were incubated sequentially with 0.02 mg/mL streptavidin and 40 pM biotin-labeled cyanine fluorophore DNA. To prevent photobleaching and extend dye lifespan, our imaging buffer contained 2 mM UV-treated trolox, 30 U/mL glucose oxidase (Sigma-Aldrich), 30 U/mL catalase (Sigma-Aldrich), and 4 mg/mL glucose. smFRET images were collected using an EMCCD (ProEM 512B, Princeton Instruments). Single cyanine dye intensity was extracted using IDL software 8.3 (ITT Vis) and analyzed using a custom-written MATLAB (r2016a) code (The MathWorks Inc.).

DNA-binding affinity assays were carried out in imaging buffer A (25 mM Tris-HCl pH 7.5, 0.5 mM EDTA, and 10% glycerol) with 50 mM or 150 mM KCl using 35-nt ssDNA overhang substrate (Oligo 5 + 6). The indicated amount of CST (0.1, 0.25, 0.5, 1, or 2 nM) or of RPA (0.05, 0.1, 0.15, 0.2, or 0.5 nM) was added and incubated for 10 min to achieve equilibrium. By combining all smFRET time traces, smFRET histograms were fitted by two Gaussians (a DNA-only state and a protein-bound state). The protein-bound fraction was analyzed by a Hill plot and Kd values were calculated in GraphPad Prism.

To assess if CST coexists with RPA on RPA-bound ssDNA, we used imaging buffer A with 150 mM KCl in the presence of the dT (13 + 47) ssDNA overhang substrate (Oligos 7 + 8). We incubated 40 nM RPA on slides for 5 min, washed with imaging buffer A with 150 mM KCl, and then added the indicated amount of CST for another 5 min incubation.

To assess RAD51 formation on CST-bound ssDNA, buffer B (25 mM Tris-HCl pH 7.5, 150 mM KCl, 2.5 mM MgCl2, 2 mM ATP) and 35-nt ssDNA overhang substrate (Oligos 5 + 6) were used. We incubated 40 nM RPA or CST on slides for 5 min, washed with imaging buffer B, and then added 800 nM RAD51 for another 5 min incubation.

**Colocalization single-molecule spectroscopy (CoSMoS)**. The CoSMoS experiment was conducted using a home-built multi-wavelength micro-mirror TIRF microscope (mmTIRFM) system according to published protocol[67]. In brief, we assembled our mmTIRF system based on a commercial framework (Mad City Labs Inc.) and installed a 100x objective (UAPON 100XOTIRF, Olympus) on the platform. Three lasers of wavelength of 488, 532, and 638 nm were expanded separately through Keplerian beam expanders and combined into the same optical path before laser light passed through the objective. The emission images were passed through a filter-wheel (Lambda 10-B, Sutter) where the specific wavelength band-pass filters were installed according to the desired fluorophore to be excited (FF01-517/20, FF01-572/15, and FF01-676/29 band-pass filters (Semrock) for the blue, green and red channels, respectively). Images were acquired using an EMCCD (iXon Ultra 897, Andor) controlled by codes written in LabView (2016) (National Instruments Corp.) by Jeff Gelles lab (https://github.com/gelles-brandeis/Glimpse).

CoSMoS experiments were carried out in buffer A with 50 mM KCl in the presence of the Cy3-labeled 80-nt ssDNA overhang substrate (Oligos 9 + 10). We incubated 10 nM EGFP-RPA on slides for 5 min, washed with imaging buffer A, and then added the indicated amount of SNAP-CST for another 5-min incubation. To determine the proportion of RPA-CST colocalization, we recorded the reaction as a 10 s movie with 1 s time exposure. Using Imscroll code[68] written in MATLAB, colocalization is scored when EGFP-RPA or SNAP₆₄₉-CST occurred within 2.2 pixels of Cy3-labeled ssDNA spots.

**Single-molecule fluorescence photobleaching experiment**. The EGFP-RPA photobleaching experiment was conducted using the 80-nt ssDNA substrate described for smFRET experiments and 10 nM EGFP-RPA. For experiments

including CST, 100 nM CST in imaging buffer A with 50 mM KCl was added and incubated for 5 min. Photobleaching was achieved using a 6–9 mW 488 nm laser, with photobleaching taking place stochastically within 2–3 min. The photobleaching time-course was recorded across at least 300 frames with an exposure time of 0.5 s to obtain an accurate time-point for the drop in signal intensity. Signal intensity data were smoothed using a two-frame average before fitting for drops in intensity state. To identify the intensity drop of EGFP-RPA fluorescence, the built-in MATLAB function "findchangepts" was used and confirmed by visual inspection.

**In vitro fluorescence labeling of SNAP-CST protein**. We incubated 5 μM SNAP-CST with 10 μM SNAP-surface-649 (New England Biolabs) in 50 μL buffer C (35 mM Tris-HCl pH 7.5, 150 mM KCl, 10% glycerol, 0.01% Igepal, 1 mM 2-mercaptoethanol) for 16 h at 4 °C. Then we used an Amicon Ultra-4 10K centrifugal tube (Millipore) to remove the excess fluorescent dye and to concentrate the labeled CST complex. The labeled CST complex was divided into small aliquots and stored at −80 °C.

**DNA pulldown assay**. We incubated 2 μL of 80-mer ssDNA-conjugated magnetic beads with 1.25 μM human RPA or CST in 10 μL of buffer D (35 mM Tris-HCl pH 7.5, 10% glycerol, 0.01% Igepal, and 1 mM 2-mercaptoethanol) containing 1 mM ATP, 2.5 mM MgCl$_2$, and 50 mM or 150 mM KCl, for 5 min at 37 °C. Indicated amounts of human CST, CTC1Δ700N-ST, RecA, or RAD51 were then added for 5-min incubation. The beads were captured using a magnetic beads separator and the unbound fractions were kept for further analysis. After washing the beads with 100 μL buffer D with 50 mM or 150 mM KCl, bound proteins were eluted in 15 μL SDS sample buffer. The unbound and bound fractions were analyzed by SDS-PAGE and Coomassie blue staining to determine the protein contents on the ssDNA by means of a Gel Doc XR+ system with Image Lab software 6.0 (Bio-Rad) and quantified using Image J 1.52a software.

**Affinity pulldown assay**. To determine direct physical protein-protein interactions, 1 μM of human CST or CTC1Δ700N-ST containing a His$_6$ tag at the C-terminus of TEN1 was incubated for 20 min at 37 °C with 1 μM RAD51, RecA, or RPA in 10 μL of buffer D with 5 mM imidazole and 50 mM KCl. To determine the interaction between CST and RPA in the presence of ssDNA, 1 μM of human CST was incubated with 1 μM RPA in 10 μL of buffer D with 5 mM imidazole, 50 mM KCl, and 2 μM 30-nt ssDNA (Oligo 11). The sample was then mixed with 2 μL of His-Tag Dynabeads (Invitrogen) for another 20 min at 37 °C to capture CST and associated proteins. The beads were captured using a magnetic beads separator and the supernatants were kept for further analysis. After washing the beads with 100 μL buffer D with 50 mM KCl, bound proteins were eluted in 15 μL SDS sample buffer. The supernatants and eluates were analyzed by SDS-PAGE and Coomassie blue staining to determine protein contents by means of a Gel Doc XR+ system with Image Lab software 6.0 (Bio-Rad).

For the interaction between RecA and CST, 0.25 μM of human CST that containing a Flag tag at the N-terminus of CTC1 was incubated for 20 min at 37 °C with 0.25 μM His-tagged RecA in 10 μL of buffer D with 50 mM KCl. The sample was then mixed with 5 μL of Anti-FLAG M2 affinity gel (Sigma-Aldrich) for another 20 min at 37 °C to capture CST and associated proteins. The beads were captured using mini centrifuge and the supernatants were kept for further analysis. After washing the beads with 100 μL buffer D with 50 mM KCl, bound proteins were eluted in 10 μL buffer D with 50 mM KCl and 500 μg/ml 3× Flag peptide. The supernatants and eluates were analyzed by SDS-PAGE and following immunoblotting with anti-His antibody (Sigma-Aldrich, H1029, 1:1000). The RecA and TEN1 contents were determined using BioSpectrum 810 imaging system with VisionWorks LS Software 8.6 (UVP).

**Electrophoretic mobility shift assay (EMSA)**. Fluorescence tag-labeled ssDNA Cy3-Oligo four substrate (80 nM) was incubated at 37 °C for 5 min with the indicated amounts of CST and SNAP$_{649}$-CST or RPA and EGFP-RPA in 10 μL buffer E (35 mM Tris-HCl pH 7.5, 1 mM DTT, 2.5 mM MgCl$_2$, and 100 ng/μL BSA) containing 50 mM KCl. The reaction mixtures were then electrophoresed on a 0.8% agarose gel in 1× TBE buffer (89 mM Tris, 89 mM borate, and 2 mM EDTA, pH 8) at 100 V for 30 min at 4 °C. Gels were analyzed in an Amersham Typhoon 5 Biomolecule imager (Cytiva) with Amersham Typhoon 2.0 software to detect Cy3 fluorescence signal.

**DNA strand-exchange assay**. The 80-mer ssDNA Oligo 1 (3 μM nucleotides) was first incubated with indicated amounts of human RPA or CST in 10 μL buffer E containing the indicated concentrations of KCl, 1 mM ATP and 5 mM CaCl$_2$, or 1 mM AMPPNP for 5 min at 37 °C. Next, we added 1 μM human RAD51 for 5-min incubation. The reaction was initiated by adding isotope-labeled 40 basepair DNA duplex (3 μM nucleotides) in 1 μL. After 20 min incubation, we added a 2.5 μL aliquot of stop buffer (240 mM EDTA, 0.5% SDS, and 3.2 μg proteinase K) to the reaction and incubated it for a further 15 min at 37 °C. The reaction mixtures were resolved in a 10% polyacrylamide gel in TBE buffer. The gel was dried onto DE81 paper (Whatman) and subjected to phosphorimaging analysis in a Personal FX phosphorimager using Quantity One software 4.6.9 (Bio-Rad).

**D-loop formation assay**. The $^{32}$P-labeled 90-mer ssDNA Oligo 12 (3 μM nucleotides) was first incubated with RPA (0.2 μM) in buffer E containing 150 mM KCl, 1 mM ATP, and 10 mM CaCl$_2$ for 5 min at 37 °C. Then, indicated amounts of CST were added for 5-min incubation. Next, we added 1 μM RAD51 for a further 5-min incubation. The reaction was initiated by adding 1 μL pBluescript plasmid dsDNA (900 μM base pairs) to a final 10 μL reaction volume. After 10-min incubation, we added a 2.5-μL aliquot of stop buffer (240 mM EDTA, 0.5% SDS, and 3.2 μg proteinase K) to the reaction and incubated it for a further 15 min at 37 °C. The reaction mixtures were resolved in a 10% polyacrylamide gel in TBE buffer. The gel was dried onto DE81 paper (Whatman) and subjected to phosphorimaging analysis in a Personal FX phosphorimager using Quantity One software 4.6.9 (Bio-Rad).

**Electron microscopy of RAD51 presynaptic filament**. The 80-mer ssDNA Oligo 1 (3 μM nucleotides) was first incubated with 200 nM human RPA or CST in 10 μL of buffer E containing 150 mM KCl, 1 mM ATP and 5 mM CaCl$_2$ for 5 min at 37 °C. Then, 1 μM human RAD51 was added for a further 10-min incubation. After filaments had formed, a 4 μL aliquot of reaction mixture was applied onto 400-mesh grids coated with fresh carbon film that had been glow-discharged. Samples were stained with 2% uranyl acetate for 1 min and examined under a Hitachi H-7100 transmission electron microscope operated at 75 keV in conjunction with a CCD camera (Gatan Model 782 Erlangshen ES500W) at a nominal magnification of 80,000x. RAD51 filament length was quantified in Image J 1.52a software.

**Co-immunoprecipitation (Co-IP)**. Co-IP was carried out as described previously[31]. In brief, HEK293T cells co-transfected with Flag-CTC1, Myc-STN1 and HA-TEN1 were lysed in lysis buffer (0.1% NP-40, 50 mM Tris-HCl pH 7.4, 50 mM NaCl, 2 mM DTT) supplemented with protease inhibitor cocktail (1 mM AEBSF, 0.3 μM aprotinin, 50 μM bestatin, 10 μM E-64, 10 μM leupeptin, 5 μM pepstain and 1 mM PMSF), sonicated on ice, and then centrifuged (17,000 × g, 15 min, 4 °C). The supernatants were precleared, treated with or without benzonase (NEB, 0.09 U/μl) for 2 h, followed by incubation with anti-Myc antibody (Santa Cruz Biotech, sc-40) overnight at 4 °C with constant rotation. An aliquot of the treated lysates (10 μl) was loaded onto a 1% DNA agarose gel and stained with ethidium bromide to assess DNA removal. Protein A agarose beads (Roche) were used for pulldown according to the manufacturer's protocol. Precipitates were washed four times with cold lysis buffer, then resuspended in lysis buffer with SDS sample loading buffer, boiled for 10 min, and immediately subjected to SDS-PAGE for immunoblotting. Images were obtained using ImageQuant LAS4000 (GE). Anti-flag (Sigma, F7425, 1:2000), anti-Myc (A-14) (Santa Cruz, sc-789, 1:500), and anti-HA (Abcam, ab13834, 1:2000) antibodies were used for detecting Flag-CTC1, Myc-STN1, and HA-TEN1, respectively. Anti-RPA70 antibody (Bethyl, A300-241A, 1:5000) was used to detect endogenous RPA.

**Immunostaining**. Following co-transfection of Flag-CTC1, Myc-STN1, and HA-TEN1 into HeLa cells, the cells were grown for 48 h on chamber slides containing media to which BrdU (20 μM) had been added. Cells were then treated with hydroxyurea (2 mM) for 6 h, fixed with 4% paraformaldehyde for 15 min, and permeabilized with 0.15% Triton X-100 for a further 15 min. After washing three times with PBS, the fixed cells were blocked with 5% BSA at 37 °C for 1 h in a humidified chamber, and then co-incubated overnight at 4 °C with anti-BrdU (Abcam, ab6326, 1:5000), anti-Flag M2 (Sigma-Aldrich, F1804, 1:500), and anti-RPA32 pS33 (Bethyl, A300-246A, 1:5000) antibodies. After washing three times with PBS, the samples were incubated with secondary antibodies (LifeTechnologies Alexa 488 anti-rat IgG, A11006, 1:500; ThermoFisher DyLight 550 anti-mouse IgG, 84540, 1:1000; DyLight 649-anti-rabbit IgG, 35565, 1:1000) at room temperature for 1 h. Slides were then washed three times with PBS and dried via a cold ethanol series, before mounting with DAPI-containing mounting medium (Vector Laboratories). Z-stack images were acquired at a thickness of 0.3-μm per slice under a Zeiss AxioImager M2 epifluorescence microscope with a 100x objective and AxioVision software. Single representative Z-slice images were selected.

**Proximity ligation assay (PLA)**. Cells grown on chamber slides were treated with 4 mM HU for 3 h, followed by pre-permeabilization with 0.25% Triton X-100 in PBS for 2 min at room temperature prior to fixation in 2% paraformaldehyde for 15 min. Slides were then washed three times (5 min each time) with PBS in Coplin jars, permeabilized with 0.25% Triton X-100 in PBS for 15 min at room temperature, and PLA was performed using the Duolink™ In Situ Detection kit (Millipore Sigma, DUO92008) following the manufacturer's protocol with minor modifications. Briefly, slides were washed three times (5 min each time) with PBS and then blocked with blocking buffer (Sigma-Aldrich, DUO82007) at 37 °C for 1 h. Primary antibodies (anti-STN1 1:100, Abcam, ab251856; anti-CTC1, 1:100, Abcam, ab230538; anti-RPA 1:200, Abcam, ab2175) were diluted in blocking buffer, dispensed onto slides, and incubated overnight at 4 °C in a humidified chamber. The slides were then washed three times (5 min each time) with wash buffer F (0.01 M Tris, 0.15 M NaCl and 0.05% Tween-20, pH 7.4), before incubating with Duolink In Situ PLA probe anti-mouse plus (Sigma-Aldrich, DUO82001) and anti-rabbit minus (Sigma-Aldrich, DUO82005) for 1 h at 37 °C.

After washing three times (5 min each time) with wash buffer F, the slides were incubated with Duolink ligation mix at 37 °C for 30 min, washed twice with wash buffer F (2 min each time), and then incubated with diluted Duolink amplification mix at 37 °C for 100 min. The slides were then washed three times (10 min each time) with wash buffer G (0.2 M Tris and 0.1 M NaCl) and subjected to a final wash in 0.01x diluted wash Buffer G for 1 min before being allowed to dry in the dark. Images were obtained using a Zeiss AxioImager M2 epifluorescence microscope after counter-staining nuclei with DAPI mounting medium. A ×20 objective was used for quantitation and a 40x objective was used to obtain representative images. Quantitation was determined by counting PLA foci with the ZEN 3.0 software (blue edition). Data were plotted using Graphpad Prism 9.0.2 software and are shown as mean ± SEM.

**Statistics**. All statistical analyses were conducted in GraphPad Prism 7 or 9.0.2 software as indicated to establish statistical significance. A D'Agostino-Pearson test or Shapiro–Wilk test was used to assess if the data were normally distributed, in which case a one-way ANOVA with Tukey's post hoc test was used to compare across multiple groups. If a sample was not normally distributed, we applied a Kruskal–Wallis test followed by Dunn's test. Unpaired two-tailed Student's $t$ tests were utilized to compare between two groups, adopting a confidence level of 95%. Unpaired two-tailed Student's $t$ tests with correction for multiple comparison using the Holm–Sidak method were used to compare differences among groups, with $\alpha = 0.05$. A $P$ value <0.05 was considered significant. Exact P values are provided in the Source Data. No statistical methods were used to pre-determine the sample size.

**Reporting summary**. Further information on research design is available in the Nature Research Reporting Summary linked to this article.

## Data availability

The data supporting the findings of this study are available from the corresponding authors upon reasonable request. Source data are provided with this paper.

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

## Acknowledgements

We thank Liuh-Yow Chen (Academia Sinica, Taiwan) for the CST constructs, Eric Greene for the GFP-RPA expression plasmid, and Jeff Gelles and colleagues (Brandeis) for providing the imaging analysis package and suggestions on microscope setup. This work was supported by Academia Sinica (P.C.), National Taiwan University (P.C. and H.-W.L.), Taiwan Ministry of Science and Technology (MOST 108-2321-B-002-054 to P.C., MOST 107-2113-M-002-010-MY3 to H.-W.L.), and NIH R01CA234266 to W.C.

## Author contributions

K.-H.L., H.-L.Y., H.-W.L., and P.C. conceived the study. K.-H.L., H.-L.Y., W.C., H.-W.L., and P.C. designed the experiments. K.-H.L. preformed the majority of protein purification and biochemical assay. H.-L.Y. preformed the single-molecule experiments and analyzed data. D.D.N. performed PLA and analyzed the data. H.-Y.C. purified the RPA and GFP-RPA proteins. H.-Y.Y. performed the EM experiment. T.-Y.L. setup the instrument of CoSMoS assay. X.L. performed the co-IP experiment and M.C. performed RPA/CTC1/BrdU immunofluorescence staining. Statistical analysis provided by each author. The paper was written by K.-H.L. and P.C. with contributions from H.-L.Y., H.-W.L., and W.C.

## Competing interests

The authors declare no competing interests.

## Additional information

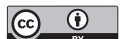

