## [Peer Review File · Nature Communications]

Crosstalk between CST and RPA regulates RAD51 activity during replication stressREVIEWER COMMENTS

Reviewer #1 (Remarks to the Author):

The manuscript by Lei and colleagues focuses on characterization of the effect of CST and RPA on RAD51 filament formation. This work is logical continuation of previous report from subset of authors (Chastain et al, 2016, Cell Reports) that demonstrated enrichment of RAD51 at telomeric and other non-telomeric GC-rich repetitive fragile sites in CST-dependent manner in response to replication stress. Here the authors show that CST and RPA can co-occupy the same ssDNA molecule and that CST physically interacts with RAD51 and recruits it to RPA-bound ssDNA via a facilitated dissociation mechanism. However, this recruitment to RPA-coated ssDNA does not promote RAD51-mediated strand exchange in contrast to CST-coated ssDNA.

This is very well and carefully performed study and clearly written text. While this work represents very interesting view into the mechanism of RAD51 recruitment/filament regulation at stalled replication forks by crosstalk between CST and RPA, further experiments are required to support not only the proposed mechanistic view but also its biological relevance.

Major suggestions:

- Suggested CST recruitment of RAD51 to RPA-bound ssDNA could imply its role in protection and restart of stalled replication forks. Therefore, such experiment in comparison with BRCA2 deficiency using fiber assay (ideally combined with telomeric signal) would be essential to provide the biological context.
- Identified interaction between CST and RAD51 should be mapped and ideally interaction deficient mutant generated that would serve as great control not only for in vitro studies but also biological relevance of this interaction. The assembly of RAD51 filament on CST-bound DNA seems contains also DNA only step (Fig. 6d), questioning the biological importance of the CST-RAD51 interaction for RAD51 filament assembly.
- Given the CST-dependent enrichment of RAD51 on telomeric and non-telomeric GC-rich sequences, how this mechanistic study would apply to G4 containing sequences?

Minor points:

- Can authors exclude that the direct in vitro CST-RPA interaction can be mediated by one or the other protein in DNA bound form? Pull-down in the presence of minimal binding size ssDNA should help address this point.
- Suggested co-occupancy of ssDNA by CST and RPA complexes would be supported by available EM analysis. In addition, does higher ionic strength (150 mM salt) affect the co-existence of RPA with CST on the same ssDNA (Fig.2c)? Similarly, is RPA also able to bind preformed CST-ssDNA complex?
- Can authors comment on role of newly observed DNA-dependent CST association with RPA that does not seem to be HU regulated?
- The effect of salt on smFRET peak of CST-bound ssDNA should be provided in supplement to better understand the possible changes in the binding mode of CST.
- The capture assay (Fig 2C) shows possible 1:1 stoichiometry of CST/RPA occupancy. This does not seem to be the case in smFRET and CoSMoS analysis (Fig 3). Can authors discuss the differences?
- It would be interesting to know the effect of the CST-RAD51 interaction on oligomeric state of these proteins given their apparent 1:1 stoichiometry in pull-down assay. Furthermore, given this stoichiometry, why only half of the molecules show smFRET reduction of CST-bound ssDNA upon addition of 20-fold excess of RAD51 (Fig. 6c)?

Reviewer #2 (Remarks to the Author):

In the manuscript by Lei et al, the authors examined the interaction between CST with RPA and

RAD51. Using immunofluorescence, biochemical, and biophysical approaches, the authors found that CST and RPA can simultaneously bind ssDNA. The authors went on to demonstrate that CST enables RAD51 loading on RPA-coated ssDNA through CTC1 DNA binding activity. Although CST can recruit and enhance RAD51 protein amounts on ssDNA and increase filament length, the authors claim that CST is paradoxically not a recombination mediator. However, more extensive RAD51 and RPA protein concentrations can be assayed to adequately address this phenomenon. Overall, the data are technically beautifully presented. The experiments are well described for a general audience. Although it was known previously that CST interacts with RAD51, this paper fills in some mechanistic gaps in our knowledge about this interaction and its purpose.

Major comments:

1) In Figure 5, the strand exchange reactions were performed with small linear substrates. It is possible that a stimulatory effect of CST on RPA-coated ssDNA was not observed due to the substrates utilized. It may be possible that substrates containing supercoiled DNA or long ssDNA substrate could alleviate this issue. A strand exchange assay with phiX (see Subramanayam S. et al., 2016 PNAS PMID: 27671650, Chen R. et al., 2016 NAR PMID: 27131385) or D-loop formation assays using pBluescript KG substrates (Raynard S. et al., 2009 CSHP PMID: 20147015 from P. Sung lab) could be assessed and may yield a stimulatory effect.

2) In Supplemental Figure 6, the RPA concentrations in the reactions were in excess (2uM), which inhibits the reaction and may prevent CST function as a recombination mediator. This experiment should be performed with lower RPA concentrations (0.1 uM, 0.050 uM, and 0.010 uM RPA) to definitively test the hypothesis that CST is not a recombination mediator. These manuscripts may be helpful, Shim, Kang-Sup et al., DNA Repair 2006; PMID: 16624636) and others from Dr. Richard Fishel lab.

3) In Figure 6b, please add the values of the length of the RAD51 filaments with their associated error.

Minor comments:

1) Fig. 3e axis should read "number" not "nubmer".

2) Abstract Line 32-33 and Line 236. RAD51 can displace RPA at higher concentrations (see Subramanayam S. et al., 2016 PNAS PMID: 27671650, Chen R. et al., 2016 NAR PMID: 27131385 etc.). This phenomenon should be more accurately described.

3) Line 56 please cite Reference 22.

4) Line 59: Note that RAD51 protein forms a right-handed nucleoprotein filament in both the absence and presence of ssDNA.

Reviewer #3 (Remarks to the Author):

This is a nice paper examine the molecular mechanisms of CST during replication fork reversal. The CST complex is emerging as an important contributing factor to the maintenance of genome integrity, including identified roles in telomere regulation and more recently identified roles in replication fork reversal. This latter phenomena is extremely important to genome integrity, but it is also relatively poorly understood. The finding that CST can now be classified as a potential regulatory of fork reversal is very exciting for the field. Here, the authors use a combination of cellular imaging, biochemical and biophysical measurements to probe the interactions of CST with RPA, RAD5 and ssDNA. The most

striking findings are that CST and RPA can co-occupy ssDNA, and CST interacts directly with RAD51 to recruit it to DNA, but surprisingly, CST itself is not a mediator (i.e. it does not help RAD51 assemble on RPA-coated DNA similar to what BRCA2 might do). These findings are important to the field and will certainly lead to further studies to help more fully define CST functions (in particular, in combination with BRCA2) - the next level of studies that will be made possible due to this paper should be very exciting. The work is all technically sound and paper is written exceptionally well and should be accessible to a range of audiences.

I have some very minor comments:

1. The PLA assay should be described in more detail to help readers unfamiliar with this technique to interpret the presented data.

2. Much of the ssDNA binding data is presented in the form of smFRET experiments. In my view, the authors findings on CST are very exciting and will garner interest from many readers without expertise in smFRET. I would strongly encourage the authors to take 2-3 sentences in the subsection of the manuscript entitled "DNA-binding properties of the CST and RPA complexes" and more carefully explain the smFRET assay and how one interprets the data (with the non expert in mind). Addition of these sentences should be most helpful for reaching a broad audience.

3. Lines 358-360 of the discussion: "...these results indicate that CST OB-fold domains with the highest ssDNA-binding affinity can compete with RPA OB-fold domains with lower..." In general, I agree with the authors' conclusion here, but I would suggest that they replace the word "indicates" with "suggests" or edit the sentence in some other way to tone down the conclusion. My complaint is that the authors have not actually done the full set of experiments (for example, tested OB-fold mutants, etc) to make such a strong conclusive statement.

4. Figure 2a and 2b - you might consider selecting a different color for the protein. For some reason, the current "peach color" shows up on my computer screen but was almost invisible in my print copy. All of the other figures / colors look fine.

Response to Reviewers Comments

We are very grateful to the three reviewers for spending time scrutinizing our study and manuscript. As detailed below, we have addressed all the issues raised by the reviewers.

Reviewer #1 (Remarks to the Author):

The manuscript by Lei and colleagues focuses on characterization of the effect of CST and RPA on RAD51 filament formation. This work is logical continuation of previous report from subset of authors (Chastain et al, 2016, Cell Reports) that demonstrated enrichment of RAD51 at telomeric and other non-telomeric GC-rich repetitive fragile sites in CST-dependent manner in response to replication stress. Here the authors show that CST and RPA can co-occupy the same ssDNA molecule and that CST physically interacts with RAD51 and recruits it to RPA-bound ssDNA via a facilitated dissociation mechanism. However, this recruitment to RPA-coated ssDNA does not promote RAD51-mediated strand exchange in contrast to CST-coated ssDNA.

This is very well and carefully performed study and clearly written text. While this work represents very interesting view into the mechanism of RAD51 recruitment/filament regulation at stalled replication forks by crosstalk between CST and RPA, further experiments are required to support not only the proposed mechanistic view but also its biological relevance.

Major suggestions:

1. Suggested CST recruitment of RAD51 to RPA-bound ssDNA could imply its role in protection and restart of stalled replication forks. Therefore, such experiment in comparison with BRCA2 deficiency using fiber assay (ideally combined with telomeric signal) would be essential to provide the biological context.

(Author response) We thank the reviewer for the comment regarding the biological function of CST in response to replication stress. Previous works by us and others have documented that the CST complex plays a critical role in reinitiating stalled DNA synthesis at both telomeric and non-telomeric sequences (Chastain et al., Cell Reports, 2016; Gu et al., EMBO J., 2012; Huang et al., Cell Res., 2012; Stewart et al., EMBO J., 2012; Wang et al., Cell Cycle, 2014). We have previously addressed whether the CST complex participates in stalled fork protection using a DNA fiber assay, as suggested by this reviewer (Lyu et al., EMBO J., 2020). Our published work clearly evidenced that depletion of STN1 or CTC1 enhanced fork degradation, and that RNAi-resistant wild-type STN1 or CTC1, respectively, rescued that phenotype (see Figs. 2D and S2B in Lyu et al., EMBO J., 2020). Collectively, the CST complex plays an important role in protecting and restarting stalled replication forks. Importantly, depletion of CST resulted in genome instabilities similar to those displayed by BRCA2-depleted cells (Lyu et al., EMBO J., 2020; Schlacher et al., Cell, 2011). We have now elaborated on these points in the revised manuscript (lines 88-96).

Given that CST and BRCA2 exhibit similar phenotypes under replication stress, our previous study also addressed their genetic relationship in response to replication stress (Lyu et al., EMBO J., 2020). Our results showed that CST depletion elevated genome instability in BRCA2-deficient cells, as revealed by various assays including detection of micronuclei and anaphase bridge, γ H2AX foci, BrdU incorporation signal, and chromosomal aberrations. We then made repeated attempts to

determine nascent strand degradation in double knock-down cells using the DNA fiber assay. However, efficient co-depletion of STN1 and BRCA2 resulted in massive cell detachment from dishes and apoptosis, and the remaining cells had abolished STN1 knockdown (please see inline **Fig. R1**; Lyu et al., EMBO J., 2020). This prevented us from obtaining data that truly represent fork dynamics in double knock-down cells. In conclusion, our data suggest an additive effect of CST and BRCA2 in response to replication stress. Consistent with cellular observations, our mechanistic findings presented in this study fully support the additive effect of CST and BRCA2 in regulating RAD51-mediated reversed fork protection. We have now elaborated on this point in our Discussion of the revised manuscript (lines 438-442).

Figure R1. Depletion of both STN1 and BRCA2 induced massive cell detachment from dishes, which prevented further DNA fiber analyses. The remaining cells were stained with crystal violet (Lyu et al., EMBO J., 2020).

2. Identified interaction between CST and RAD51 should be mapped and ideally interaction deficient mutant generated that would serve as great control not only for in vitro studies but also biological relevance of this interaction. The assembly of RAD51 filament on CST-bound DNA seems contains also DNA only step (Fig. 6d), questioning the biological importance of the CST-RAD51 interaction for RAD51 filament assembly.

(Author response) We appreciate the reviewer’s suggestion. CST forms a complex, binds DNA, and interacts with RAD51. Our previous work had attempted to identify separation-of-function CST variants defective in RAD51 interaction. We generated 11 clinically-relevant missense and small-deletion pathogenic CTC1 mutant variants to characterize their functional interaction with RAD51 (Wang and Chai, Nucleic Acids Res., 2018). Among them, the CTC1 R975G and CTC1 C985Δ mutants retained CST complex formation but lacked the interaction with RAD51 (Wang and Chai, Nucleic Acids Res., 2018). Although both variants exhibited global genome instabilities, which were further elevated by replication stress, purified CTC1 R975G and CTC1 C985Δ mutant proteins also exhibited a lack of DNA-binding activity (Chen et al., Genes Dev., 2013). As such, it is difficult to identify distinct separation-of-function CST variants specifically defective in RAD51 interaction. We now elaborate on this point in in the revised manuscript (lines 404-420). To address the reviewer’s concern regarding the specificity of functional interaction between CST and RAD51 *in vitro*, we have now assessed if CST displays a functional interaction with the prokaryotic recombinase RecA. We found that CST does not interact with RecA (**Supplementary Fig 5a**) and, most importantly, RecA cannot be recruited to the RPA-coated ssDNA by CST (new data in **Supplementary Fig 5b**, lines 267-268, also see inline figure for the reviewer’s convenience). This result supports the notion that physical interaction is crucial for CST to recruit RAD51 to RPA-coated ssDNA.

Regarding Fig. 6d, we disagree that the assembly of RAD51 filament on CST-bound DNA seems to involve a DNA-only step. FRET is a technique sensitive to the distance between two dyes. The high FRET intermediate (~0.6) observed in our real-time RAD51 assembly experiment shown in Fig. 6d reflects the change in dye-pair separation from the “CST fully-bound state (0.2)” and the “RAD51

assembled state (0.05)". This high-FRET intermediate could have resulted from either partial or transient dissociation of CST from ssDNA. In fact, our results show that RAD51 can efficiently assemble on CST-coated ssDNA, but not on RPA-coated ssDNA. Whether RAD51 is associated with CST during the CST-bound state or the partial/transient CST dissociation state cannot be concluded from our FRET experiments.

In conclusion, our previous cell-based and ChIP studies clearly demonstrate that CST deficiency significantly attenuates RAD51 foci formation and recruitment to fragile sequences under replication stress (Chastain et al., Cell Reports, 2016; Wang and Chai, Nucleic Acids Res., 2018). In the current study, our *in vitro* analysis directly evidences physical and functional interactions between CST and RAD51, but not RecA. Thus, CST plays an important role in efficient RAD51 recruitment to damaged sites upon fork stalling. It is worth emphasizing that loss of CST DNA-binding activity significantly attenuated HU-induced RAD51 foci formation (Lyu et al., EMBO J., 2020), indicating that the DNA-binding activity of CST is also a prerequisite for efficient RAD51 recruitment to stalled forks. We have now elaborated on the attributes of CST DNA-binding and protein interactions for RAD51 recruitment in response to replication stress in the revised manuscript (lines 404-420).

(new data) now Supplementary Fig 5b. For ssDNA pulldown analysis, RPA was preincubated with magnetic ssDNA beads. Then, CST and RAD51 or RecA were added to complete the reaction under the condition of 50 mM KCl. The unbound and bound fractions from the reaction were analyzed by 15% SDS-PAGE with Coomassie blue staining.

3. Given the CST-dependent enrichment of RAD51 on telomeric and non-telomeric GC-rich sequences, how this mechanistic study would apply to G4 containing sequences?

(Author response) Thank you for this interesting point. Our previous ChIP analyses have shown that CST deficiency significantly reduces RAD51 recruitment to telomeric and non-telomeric GC-rich

sequences under replication stress (Chastain et al., Cell Reports, 2016). As the reviewer mentions, single-strand G-rich repetitive sequences are prone to forming G-quadruplex structures, suggesting a potential function for CST-RAD51 in G4 secondary structures. Recent cellular and single-molecule studies have documented that CST can unfold G4 structure and is recruited to telomeric and non-telomeric DNA in response to G4 formation (Bhattacharjee et al., Nucleic Acids Res., 2017; Zhang et al., Nucleic Acids Res., 2019). Furthermore, CST prevents G4-induced inhibition of DNA replication, and CST depletion slows lagging-strand telomere replication after G4 stabilization (Zhang et al., Nucleic Acids Res., 2019). These data indicate that CST unfolds G4 structures to prevent or resolve replication stalling at G4 sites. We hypothesize that CST prevents the accumulation of G4 secondary structures to allow efficient binding of RAD51 to ssDNA and enable restart of the replication fork. We now elaborate on this point in the Discussion of our revised manuscript (lines 357-368).

Minor points:

4. Can authors exclude that the direct in vitro CST-RPA interaction can be mediated by one or the other protein in DNA bound form? Pull-down in the presence of minimal binding size ssDNA should help address this point.

(Author response) Thank you for this suggestion. Since the DNA binding size of RPA is about 20-30 nucleotides, we used a 30 nt-long ssDNA for the affinity pulldown assay. First, we show by gel shift assay that both CST and RPA could strongly bind the 30 nt ssDNA (**inline Fig. R2**). Second, and most importantly, we observed no physical interaction between CST and RPA in the presence of ssDNA. Therefore, we can exclude that the possible interaction between CST and RPA is mediated by ssDNA. We have now included this pulldown assay in the revised manuscript (new **Fig. 2d**, lines 184-186, also see inline figure for reviewer's convenience).

Figure R2. The DNA-binding ability of CST and RPA with 100 nM 30-nt ssDNA was determined by EMSA and stained with SYBR Gold.

(new data) now Fig 2d. Lack of physical interaction between RPA and CST in the presence or absence of ssDNA.

d. Affinity pulldown assay. Flag-CTC1-STN1-TEN1-His₆ (1 μM) was incubated with RPA (1 μM) in the presence of 2 μM 30-nt ssDNA, followed by incubation with His-Tag Dynabeads to capture CST and associated proteins using a magnetic bead separator. The supernatant (S) and eluate (E) were analyzed by 15% SDS-PAGE with Coomassie blue staining. RPA alone is shown as a control.

5. Suggested co-occupancy of ssDNA by CST and RPA complexes would be supported by available EM analysis. In addition, does higher ionic strength (150 mM salt) affect the co-existence of RPA with CST on the same ssDNA (Fig.2c)? Similarly, is RPA also able to bind preformed CST-ssDNA complex?

(Author response) We agree that it would be great to observe by EM co-occupancy of ssDNA by CST and RPA. However, we could not capture distinct image features of CST or RPA under our EM conditions with negative staining to be able to distinguish them. Accordingly, we instead focused on imaging-based CoSMoS experiments to show coexistence of CST and RPA on the same ssDNA (see Fig. 3c).

To address the issue of ionic strength affecting the co-existence of RPA with CST on the same ssDNA, we now added new sets of a ssDNA pulldown analysis under various salt conditions. We found that CST could coexist with preformed RPA-ssDNA at low ionic strength (50 mM salt), but not at high ionic strength (150 mM salt; see new **Supplementary Fig. 2a**, also see inline figure for reviewer convenience). Consistent with this notion, RPA could coexist with preformed CST-ssDNA at the lower ionic strength but not at the higher ionic strength (see new **Supplementary Fig. 2b**, also see inline figure for reviewer convenience). Collectively, those data are consistent with our conclusion and reflect the fact that CST harbors a similar DNA-binding affinity as RPA under lower ionic strength. We have now included these new data in the revised manuscript (new **Supplementary Figs. 2a and 2b**, lines 178-183).

(new data) now Supplementary Figure 2

a A ssDNA pulldown assay. RPA was preincubated with magnetic ssDNA beads and then the indicated amounts of CST were added under the condition of 150 mM KCl. The unbound and bound fractions from the reaction were analyzed by 15% SDS-PAGE with Coomassie blue staining.

b CST was preincubated with magnetic ssDNA beads and then the indicated amounts of RPA were added under the condition of 50 or 150 mM KCl.

6. Can authors comment on role of newly observed DNA-dependent CST association with RPA that does not seem to be HU regulated?

(Author response) We appreciate this insightful comment. Our PLA experiments clearly evidence the co-localization of endogenous CST and RPA under HU regulation (**Fig. 1b**). The reason why we did not observe DNA-dependent CST association with RPA in response to HU treatment via immunoprecipitation (as shown in Fig. 2e) may simply be due to overexpression of the CST complex. Excess CST could promote co-existence of CST with RPA on DNA even in the condition without HU treatment. Thus, the effect of DNA-dependent CST association with RPA in response to HU could not be concluded by this assay.

7. The effect of salt on smFRET peak of CST-bound ssDNA should be provided in supplement to better understand the possible changes in the binding mode of CST.

(Author response) Thank you for this suggestion. Both RPA and CST contain multiple OB-fold domains with different DNA affinities, so their DNA-binding is likely to be salt-sensitive. As FRET measurements are sensitive to the distance between two dyes, our FRET histograms show that RPA and CST bind differentially to DNA (dT13+47) under high (150 mM) and low salt (50 mM) concentrations. At high salt, the FRET value of the RPA-bound state (0.3) is significantly different from that of the CST-bound state (0.55). As suggested by the reviewer, we now included the smFRET peak of CST- and RPA-bound ssDNA under different salt conditions in the supplementary data of the revised manuscript (new **Supplementary Fig. 1c**, lines 208-211, also see inline figure for reviewer's convenience).

(new data) now Supplementary Figure 1c

Differences in the smFRET histograms reveal the effect of salts on the DNA-binding of CST and RPA, using the dT (13+47) ssDNA overhang substrate. The smFRET histograms for CST are plotted

in gray bars (left), whereas those for RPA are in open bars (right). Both CST- and RPA-bound ssDNA display smFRET values of ~ 0.2 in 50 mM KCl (upper panel). In 150 mM KCl (lower panel), RPA-bound ssDNA retains the same smFRET value of ~ 0.2 , whereas the smFRET value of CST-bound ssDNA changes to ~ 0.55 .

8. The capture assay (Fig 2C) shows possible 1:1 stoichiometry of CST/RPA occupancy. This does not seem to be the case in smFRET and CoSMoS analysis (Fig 3). Can authors discuss the differences?

(Author response) Thank you for pointing this out. Our objective was not to define the stoichiometry of CST and RPA binding to DNA, either by means of pull-down, smFRET, or CoSMoS assays. We only endeavored to show that CST and RPA co-occupy the same DNA. As no apparent RPA-CST interaction exists either in solution or in the presence of ssDNA (also see response in #4), the relative stoichiometry is irrelevant. For example, our smFRET histograms shown in Fig. 3b illustrate an increasing population of CST-RPA co-occupied DNA when more CST is added, and our CoSMoS experiments in Fig. 3e illustrate that RPA is likely not displaced during CST-RPA co-occupancy.

9. It would be interesting to know the effect of the CST-RAD51 interaction on oligomeric state of these proteins given their apparent 1:1 stoichiometry in pull-down assay. Furthermore, given this stoichiometry, why only half of the molecules show smFRET reduction of CST-bound ssDNA upon addition of 20-fold excess of RAD51 (Fig. 6c)?

(Author response) Thank you for this suggestion regarding the stoichiometry of the CST-RAD51 interaction. Although our affinity pulldown assay results may be interpreted as showing a 1:1 stoichiometry, we are not comfortable categorically making that conclusion based on a single experimental approach. Ideally, analytical ultracentrifugation (AUC) should be deployed to precisely address this topic. Since a large quantity of purified proteins is required for this analysis and our ability to obtain sufficient amounts of highly homogeneous CST recombinant proteins is limited at present, we feel that the stoichiometry of the CST-RAD51 interaction should be addressed instead in a future study.

In our smFRET experiments, ssDNA is pre-coated with CST, and then RAD51 is introduced, so there is no pre-existing CST-RAD51 complex in solution. In fact, a micromolar concentration of RAD51 was required for efficient assembly onto bare ssDNA in our smFRET experiments. As shown in the smFRET histograms below (inline **Fig. R3**), 200 nM RAD51 cannot form nucleoprotein filaments, whereas 400 nM RAD51 could form filaments, likely dynamically.

Fig. R3. The smFRET histograms show formation of RAD51 nucleoprotein filament in the 80-nt ssDNA overhang substrate. In both 50 and 150 mM KCl, the FRET value in the presence of 200 nM RAD51 (left panel) is the same as DNA-only state (~0.55), suggestive of no nucleoprotein filament formed. In 400 nM RAD51, low FRET distribution becomes apparent, reflective of the RAD51-bound filaments.

Reviewer #2 (Remarks to the Author):

In the manuscript by Lei et al, the authors examined the interaction between CST with RPA and RAD51. Using immunofluorescence, biochemical, and biophysical approaches, the authors found that CST and RPA can simultaneously bind ssDNA. The authors went on to demonstrate that CST enables RAD51 loading on RPA-coated ssDNA through CTC1 DNA binding activity. Although CST can recruit and enhance RAD51 protein amounts on ssDNA and increase filament length, the authors claim that CST is paradoxically not a recombination mediator. However, more extensive RAD51 and RPA protein concentrations can be assayed to adequately address this phenomenon. Overall, the data are technically beautifully presented. The experiments are well described for a general audience. Although it was known previously that CST interacts with RAD51, this paper fills in some mechanistic gaps in our knowledge about this interaction and its purpose.

Major comments:

1. In Figure 5, the strand exchange reactions were performed with small linear substrates. It is possible that a stimulatory effect of CST on RPA-coated ssDNA was not observed due to the substrates utilized. It may be possible that substrates containing supercoiled DNA or long ssDNA substrate could alleviate this issue. A strand exchange assay with phiX (see Subramanayam S. et al., 2016 PNAS PMID: 27671650, Chen R. et al., 2016 NAR PMID: 27131385) or D-loop formation assays using pBluescript KG substrates (Raynard S. et al., 2009 CSHP PMID: 20147015 from P. Sung lab) could be assessed and may yield a stimulatory effect.

(Author response) We thank this reviewer for the suggestion to use an alternative strand exchange assay to further verify the mediator function of CST. As recommended, we have conducted a new D-loop assay to further explore this topic. For that assay, a ssDNA substrate was pre-incubated with RPA to form the RPA-ssDNA complex. Then, CST was added for further incubation. The D-loop reaction was then initiated by adding long supercoiled pBluescript dsDNA (see inline new

Supplementary Fig. 6d). Consistent with the observations from our oligonucleotide-based strand exchange assay, inclusion of CST in the reaction did not promote RAD51-mediated homologous DNA pairing and strand exchange activity with the preformed RPA-ssDNA substrate (see inline new **Supplementary Fig. 6e**). We have now added this new D-loop assay data in the revised manuscript (lines 335-339) and in the supplementary data (new **Supplementary Figs. 6d and e**).

(new data) now Supplementary Figs. 6d and e

d Schematic of the D-loop formation assay. The ^{32}P -labeled 90-nt ssDNA is marked by an asterisk.

e RPA-bound ssDNA was used as the substrate to monitor RAD51-mediated DNA pairing and strand-exchange activity in the presence or absence of the indicated amounts of CST under the condition of 150 mM KCl. Note that 10 mM CaCl_2 was used to stimulate RAD51 activity.

2. In Supplemental Figure 6, the RPA concentrations in the reactions were in excess (2 μM), which inhibits the reaction and may prevent CST function as a recombination mediator. This experiment should be performed with lower RPA concentrations (0.1 μM , 0.050 μM , and 0.010 μM RPA) to definitively test the hypothesis that CST is not a recombination mediator. These manuscripts may be helpful, Shim, Kang-Sup et al., DNA Repair 2006; PMID: 16624636) and others from Dr. Richard Fishel lab.

(Author response) In fact, we used a low RPA concentration (0.2 μM), not 2 μM , in our strand-exchange assay, as shown in Supplementary Fig. 6b. To further address the recombination mediator function of CST, we have now conducted both strand-exchange and D-loop assays with even lower RPA concentrations (0.05, 0.1, and 0.2 μM RPA). As shown in the figure below, there was no detectable mediator activity for CST under various RPA concentrations (new **Supplementary Figs. 6c and e**, also see inline figure for reviewer's convenience). In summary, our results show that CST, unlike BRCA2, lacks detectable recombination mediator activity under well-defined conditions. We have now clarified this point in the revised manuscript (lines 333-339) and included the new data in the supplementary information (new **Supplementary Fig. 6c**).

(new data) now Supplementary Fig. 6

a Schematic of the DNA strand-exchange assay that we used to examine the mediating activity of CST. The ³²P-labeled DNA is marked by an asterisk.

c The indicated amounts of RPA were used to form RPA-bound ssDNA as the substrate to measure RAD51-mediated strand-exchange activity in the presence of CST.

3. In Figure 6b, please add the values of the length of the RAD51 filaments with their associated error.

(Author response) We appreciate this suggestion. This figure is a box and whisker plot, with the box showing the 25th and 75th percentiles and encompassing the median, while the whiskers represent the minimum and maximum values. We have now added the median values for the length of the RAD51 filaments in Fig. 6b of the revised manuscript (see inline figure for the reviewer's convenience).

Minor comments:

4. Fig. 3e axis should read “number” not “nubmer”.

(Author response) Apologies for this error – now corrected.

5. Abstract Line 32-33 and Line 236. RAD51 can displace RPA at higher concentrations (see Subramanayam S. et al., 2016 PNAS PMID: 27671650, Chen R. et al., 2016 NAR PMID: 27131385 etc.). This phenomenon should be more accurately described.

(Author response) We appreciate this comment. Indeed, previous studies have documented that RAD51 can catalyze the preformed RPA-ssDNA substrate for strand exchange at higher RAD51 concentrations (Chen et al., NAR, 2016; Ma et al., NAR, 2017; Subramanyam et al., PNAS, 2016). To avoid misleading the readers, we now emphasize this point in the Introduction and have added these references to the revised manuscript (lines 66-67; new References # 23, 28, and 29).

6. Line 56 please cite Reference 22.

(Author response) Thank you. We now cite this reference as recommended (new line 52, new Reference # 20).

7. Line 59: Note that RAD51 protein forms a right-handed nucleoprotein filament in both the absence and presence of ssDNA.

(Author response) Thank you for this reminder. Here, we intended to emphasize that RAD51 binds ssDNA and forms a functional nucleoprotein filament for catalyzing strand exchange. To avoid misleading the readers, we have now rephrased this sentence to “In its recombination role, RAD51 forms a functional nucleoprotein filament on ssDNA that is capable of searching for and locating the homologous template to initiate repair” (new lines 55-57 in the revised manuscript).

Reviewer #3 (Remarks to the Author):

This is a nice paper examine the molecular mechanisms of CST during replication fork reversal. The CST complex is emerging as an important contributing factor to the maintenance of genome integrity, including identified roles in telomere regulation and more recently identified roles in replication fork reversal. This latter phenomenon is extremely important to genome integrity, but it is

also relatively poorly understood. The finding that CST can now be classified as a potential regulatory of fork reversal is very exciting for the field. Here, the authors use a combination of cellular imaging, biochemical and biophysical measurements to probe the interactions of CST with RPA, RAD5 and ssDNA. The most striking findings are that CST and RPA can co-occupy ssDNA, and CST interacts directly with RAD51 to recruit it to DNA, but surprisingly, CST itself is not a mediator (i.e. it does not help RAD51 assemble on RPA-coated DNA similar to what BRCA2 might do). These findings are important to the field and will certainly lead to further studies to help more fully define CST functions (in particular, in combination with BRCA2) - the next level of studies that will be made possible due to this paper should be very exciting. The work is all technically sound and paper is written exceptionally well and should be accessible to a range of audiences.

I have some very minor comments:

1. The PLA assay should be described in more detail to help readers unfamiliar with this technique to interpret the presented data.

(Author response): Thank you for this suggestion. We now include a schematic of our PLA approach, as well as a detailed description of this technique, in the revised manuscript (new **Fig. 1b (i)**, and lines 995-1002 in the revised manuscript).

“In situ proximity ligation assay (PLA) is a technique to detect the physical proximity of two different proteins. In principle, if the two proteins are less than 40 nm apart, fluorescence signal can be detected. In brief, the two target proteins are bound by specific primary antibodies. If the target proteins are sufficiently proximal, PLA secondary antibodies hosting oligonucleotide can be ligated by means of two PLUS/MINUS PLA oligos to circularize. The DNA polymerase phi29 then processes rolling-circle amplification, and the resulting copies can be detected by hybridizing the fluorescence-labeled oligonucleotide.”

2. Much of the ssDNA binding data is presented in the form of smFRET experiments. In my view, the authors findings on CST are very exciting and will garner interest from many readers without expertise in smFRET. I would strongly encourage the authors to take 2-3 sentences in the subsection of the manuscript entitled "DNA-binding properties of the CST and RPA complexes" and more carefully explain the smFRET assay and how one interprets the data (with the non expert in mind). Addition of these sentences should be most helpful for reaching a broad audience.

(Author response) We thank the reviewer for this suggestion. We have now added the following brief description:

“smFRET experiments measure the change in distance between dye pairs. Upon proteins binding to DNA, the separation between dye pairs increases, thereby diminishing smFRET values. Thus, smFRET experiments enable sensitive determination of protein binding to DNA in real-time.” (lines 158-161 in the revised manuscript).

3. Lines 358-360 of the discussion: "...these results indicate that CST OB-fold domains with the highest ssDNA-binding affinity can compete with RPA OB-fold domains with lower..." In general, I agree with the authors' conclusion here, but I would suggest that they replace the word "indicates" with "suggests" or edit the sentence in some other way to tone down the conclusion. My complaint is that the authors have not actually done the full set of experiments (for example, tested OB-fold mutants, etc) to make such a string conclusive statement.

(Author response) Acknowledged. We have now replaced the word "indicate" with "suggest" to tone down our conclusion (new lines 391-393). Thank you for this suggestion.

4. Figure 2a and 2b - you might consider selecting a different color for the protein. For some reason, the current "peach color" shows up on my computer screen but was almost invisible in my print copy. All of the other figures / colors look fine.

(Author response) We thank the reviewer for this suggestion. We have made the respective modifications to render our illustrations more easily interpretable (new Figs. 2a and b; also see inline figure for reviewer's convenience).

SUMMARY

As detailed above, we have expended considerable time and effort to address all of the reviewers' comments and to incorporate all of their suggestions. We sincerely hope that you and the three reviewers will find our responses and revisions satisfactory, and that the revised manuscript is now deemed suitable for publication in *Nature Communications*.

Sincerely,

Peter Chi, Ph.D.

Professor,
Institute of Biochemical Sciences,
National Taiwan University

REVIEWERS' COMMENTS

Reviewer #1 (Remarks to the Author):

The authors did good job in addressing most of the points I raised during the first round of review. However, in my view two issues need clarification prior publication.

The DNA-only step in RAD51 filament assembly: not clear what the authors disagree with as they use similar statement in their manuscript: "RAD51 nucleoprotein assembly on CST-coated ssDNA requires the transition to the DNA-only state" (lines 319-322, and legend of Fig 6D). Since the high FRET intermediate (~ 0.6) corresponds to DNA only signal (Fig. 3B (i)) and this step is apparent in almost all trajectories (Fig. 6d) it should be carefully explained. What would be this subpopulation in Fig. 6C, if axis expanded to higher FRET signals? Can this indicate possible transient hand over mechanism?

I still believe the paper would greatly benefit from providing the mechanistic link between CST-mediated targeting of RAD51 to RPA-coated ssDNA and observed nascent strand protection or replication restart. While it does not seem to be required to formation of strand exchange/D-loop competent RAD51 filament, it could lead to formation of filaments capable of nascent strand protection observed previously. This could be supported by CST-dependent RAD51 nucleoprotein assembly on the RPA-preformed ssDNA complexes and consequent protection assay (Lee et al 2019).

Reviewer #2 (Remarks to the Author):

The reviewers have now adequately addressed my concerns.

Reviewer #3 (Remarks to the Author):

The authors have addressed all of my previous comments and the manuscript should be well received by the DNA repair community. This is an important study and I recommend it for publication.

Response to Reviewers Comments

We are very grateful to the three reviewers for their efforts in scrutinizing our manuscript. We have now clarified the remaining concerns raised by Reviewer 1 in the revised manuscript.

Reviewer #1 (Remarks to the Author):

The authors did good job in addressing most of the points I raised during the first round of review. However, in my view two issues need clarification prior publication.

1. The DNA-only step in RAD51 filament assembly: not clear what the authors disagree with as they use similar statement in their manuscript: “RAD51 nucleoprotein assembly on CST-coated ssDNA requires the transition to the DNA-only state” (lines 319-322, and legend of Fig 6D). Since the high FRET intermediate (~0.6) corresponds to DNA only signal (Fig. 3B (i)) and this step is apparent in almost all trajectories (Fig. 6d) it should be carefully explained. What would be this subpopulation in Fig. 6C, if axis expanded to higher FRET signals? Can this indicate possible transient hand over mechanism?

(Author response)

(1) Regarding the statement of “DNA-only”, we apologize for the confusion. Both we and the reviewer are referring to the same notion, but we had not made it sufficiently clear. We have replaced the term “DNA-only” with “high-FRET intermediate”. In addition, we have added the following sentences in the revised manuscript for further clarity (lines 322-326).

“As FRET experiments describe separation between the dye pairs, the exact nature of this high-FRET state remains to be characterized. It is possible that this high-FRET intermediate state could result from partial dissociation of CST or transient interactions of CST or the CST-RAD51 complex.”

(2) This “high-FRET intermediate” subpopulation (0.65) is very minor (see inline figure), as it is rather transient.

In conclusion, this high-FRET intermediate could have arisen either from partial or transient dissociation of CST from ssDNA. Thus, whether RAD51 is associated with CST during the CST-bound state or the partial/transient CST dissociation state cannot be concluded from our FRET experiments. We acknowledge that the notion of a handover mechanism warrants further investigation, but those experiments are perhaps beyond the scope of our current study.

2. I still believe the paper would greatly benefit from providing the mechanistic link between CST-mediated targeting of RAD51 to RPA-coated ssDNA and observed nascent strand protection or replication restart. While it does not seem to be required to formation of strand exchange/D-loop competent RAD51 filament, it could lead to formation of filaments capable of nascent strand protection observed previously. This could be supported by CST-dependent RAD51 nucleoprotein assembly on the RPA-preformed ssDNA complexes and consequent protection assay (Lee et al 2019).

(Author response)

We appreciate this insightful comment. We agree that CST-mediated targeting of RAD51 to RPA-coated ssDNA could potentially lead to nascent strand protection. Whether the CST-RAD51 axis alone is sufficient for protection, or other players (such as BRCA2) are involved, is a question for further study. We now mention this possibility of the involvement of CST-RAD51 in nascent strand protection in our Discussion of the revised manuscript (lines 446-452).

Reviewer #2 (Remarks to the Author):

The reviewers have now adequately addressed my concerns.

(Author response) We thank this reviewer for her/his efforts in reviewing our manuscript.

Reviewer #3 (Remarks to the Author):

The authors have addressed all of my previous comments and the manuscript should be well received by the DNA repair community. This is an important study and I recommend it for publication.

(Author response): We appreciate this reviewer for recognizing the importance of our study.

SUMMARY

We hope you and the reviewers will now agree that we have addressed all of your concerns and incorporated all of your suggestions. We sincerely hope that our revised manuscript is now deemed suitable for publication in *Nature Communications*.

Sincerely,

Peter Chi, Ph.D.

Professor,
Institute of Biochemical Sciences,
National Taiwan University